# THE LANGUAGE OF TIME: A LANGUAGE MODEL PERSPECTIVE ON TIME SERIES FOUNDATION MODELS

## ABSTRACT

Large language models have established a successful paradigm of training foundation models on massive datasets, extending this approach to multiple domains. Time series foundation models extend this paradigm, demonstrating exceptional cross-domain transfer and prediction capabilities in both industrial and academic scenarios. This creates a paradox: while time series from different domains reflect distinct dynamical systems that should limit transferability, empirical results demonstrate remarkable cross-domain performance. To resolve this paradox, this paper investigates the representation learning mechanisms and generalization capabilities of patch-based time series foundation models from both theoretical and experimental perspectives. We demonstrate that these models fundamentally extend language model representations from deterministic vectors to probabilistic distributions, enabling effective cross-domain transfer. Our analysis shows that time series patches can be quantized into discrete vocabularies with statistical properties similar to natural language. This theoretical framework explains how time series models inherit the robust representation and transfer abilities of large language models, accounting for their superior performance in temporal tasks. Our work provides a rigorous theoretical foundation for understanding, evaluating, and improving the safety and reliability of large-scale time series foundation models for time series analysis.

## 1 INTRODUCTION

Time series data capture system dynamics through regularly sampled numerical observations, characterized by temporal dependencies, trends, and periodic patterns Montgomery et al. (2015); Shumway et al. (2000), with applications spanning traffic forecasting Zheng & Huang (2020); Xie et al. (2023), supply chain management Dantzig & Ramser (1959), climate science Fu et al. (2024), and urban mobility Barbosa et al. (2018). Foundation models have transformed this field, inspired by large language models (LLMs) Devlin et al. (2019); Brown et al. (2020), achieving unprecedented zero-shot and few-shot learning capabilities through large-scale pre-training on heterogeneous datasets Ansari et al. (2024); Goswami et al. (2024); Das et al. (2024b); Woo et al. (2024); Shi et al. (2024); Rasul et al. (2023), with superior forecasting accuracy and cross-domain generalization in both industrial and academic scenarios Supcon (2024); Ye et al. (2024); Das et al. (2024a).

**Research Gap and Core Insight.** Despite empirical advances in time series foundation models, fundamental questions regarding their theoretical foundations remain unresolved. Recent studies identify critical limitations: lack of formal theoretical frameworks explaining generalization capabilities Zhang et al. (2025), doubts about cross-domain transfer effectiveness Ding et al. (2025), and challenges in understanding in-context learning mechanisms He et al. (2023); Zhao et al. (2024). The central paradox concerns domain transfer: while time series from different domains represent distinct dynamical systems with potentially incompatible temporal patterns, empirical results demonstrate exceptional cross-domain performance, raising concerns about model reliability and absence of theoretical guarantees Moreno-Torres et al. (2012); Long et al. (2018). We address this gap by demonstrating that patch-based time series foundation models can be formally characterized as a generalization of large language models, operating not on discrete tokens but on probability distributions over temporal motifs Nie et al. (2022); Dong et al. (2023); Qiu et al. (2025).

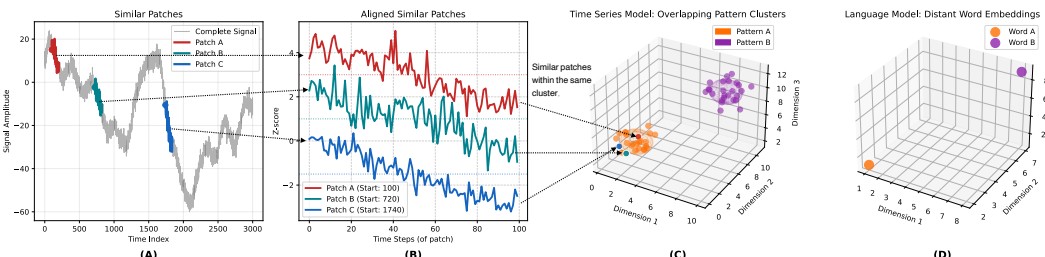

Figure 1: Temporal pattern discovery and embedding comparison. (A) Three patches (A/B/C) share identical trend shapes despite different amplitudes. (B) Normalization reveals their shared temporal motif structure. (C-D) Language tokens form discrete points in embedding space, while time series patches create probability clouds with continuous regions for the same motifs, demonstrating "distributional tokens."

**Distributional Representation Theory.** Our central theoretical contribution establishes that patch-based time series models extend the LLM paradigm from discrete token embeddings to distributional representations of temporal patterns, where patch embeddings correspond to probability distributions over families of recurring temporal motifs rather than discrete points in latent space. Unlike point-based methods Zhou et al. (2021); Wu et al. (2021) that process individual observations independently, patch-based methods Nie et al. (2022); Dong et al. (2023); Qiu et al. (2025) aggregate sequential information into temporal *motifs* Mueen (2014); Lonardi & Patel (2002), creating distributional embeddings as "probability clouds" in latent space (Figure 1). Crucially, we demonstrate that patch-based prediction fundamentally operates as a *cluster-plus-offset* mechanism: the model first classifies each patch into discrete temporal motif clusters, then predicts continuous offsets relative to the selected cluster centroid. This transforms complex temporal forecasting into a principled combination of discrete pattern recognition and continuous regression, explaining the superior performance of patch-based approaches. This paradigm shift from memorizing raw sequences to learning a distributional lexicon enables patch-based models to inherit fundamental LLM properties of abstraction, compositionality, and strong generalization Kaplan et al. (2020), providing the theoretical foundation for cross-domain transfer and robustness.

**Research Methodology.** To validate our "distributional token" hypothesis, this research integrates empirical analysis with rigorous theoretical development. We conduct comprehensive experiments using diverse real-world time series datasets to validate the key assumptions of our framework and demonstrate their practical manifestations. Concurrently, we develop a hierarchical mathematical framework that transforms intuitive insights about temporal pattern representation into formally provable conclusions, establishing both theoretical guarantees and practical guidelines for model design and evaluation.

## 2 EXPERIMENTAL EVIDENCE ON TIME SERIES DATA

Our experiments analyzed 37 diverse datasets spanning multiple domains, totaling 187 million data points and generating 790 million temporal patches with multiple patch lengths.

### 2.1 PATCHES V.S. POINTS

We conducted a comprehensive semantic analysis comparing different patch sizes, which also compare point-based methods (patch $P = 1$) with patch-based methods (patch $P = 16, 32, 64, 128, 256$). Figure 2 presents a detailed statistical evaluation across four critical dimensions. Detailed calculation and interpretation of the results can be found in the Appendix D.

Our analysis identifies four key characteristics that differentiate point-based and patch-based approaches in time series representation:

Table 1: Overview of Core Problems, Theoretical Solutions, and Key Contributions

| Core Problem | Our Solution | Key Contribution | Main Result & Impact |
|---|---|---|---|
| **Domain Transfer Paradox** Why do models transfer across distinct dynamical systems? | **Probabilistic Pattern Representation** Patches as probability distributions over recurring temporal motifs | **Time Series as Distributional Language Models** Foundation models inherit LLM's abstraction and generalization capabilities | Time series data across 37 diverse datasets present strong quasi-language properties |
| **Patch vs Point Debate** Lack of theoretical resolution for patch superiority | **Capacity Hierarchy Theory** Formal proof: point methods are special cases of patch methods ($\mathcal{H}_{point} \subseteq \mathcal{H}_{patch}$) | **Theoretical Superiority of Patches** Mathematical guarantee of patch-based representational advantages | Patch-based methods demonstrate consistently superior performance with theoretical backing for model design |
| **Expressiveness Validation** Limited empirical evidence of language-like properties | **Comprehensive Statistical Analysis** Systematic analysis of temporal pattern distributions across domains | **Language-like Statistical Laws** Discovery of Zipfian distributions in temporal vocabularies | Comprehensive evidence supporting distributional token hypothesis across 790M temporal patches |
| **Theoretical Foundation Gap** Missing mathematical framework explaining empirical success | **Unified Mathematical Framework** Cluster-plus-offset mechanism integrating discrete motifs with continuous prediction | **Complete Theoretical Foundation** Rigorous mathematical theory unifying multiple analytical perspectives | Provides principled guidelines for model design, evaluation, and reliability assessment in practice |

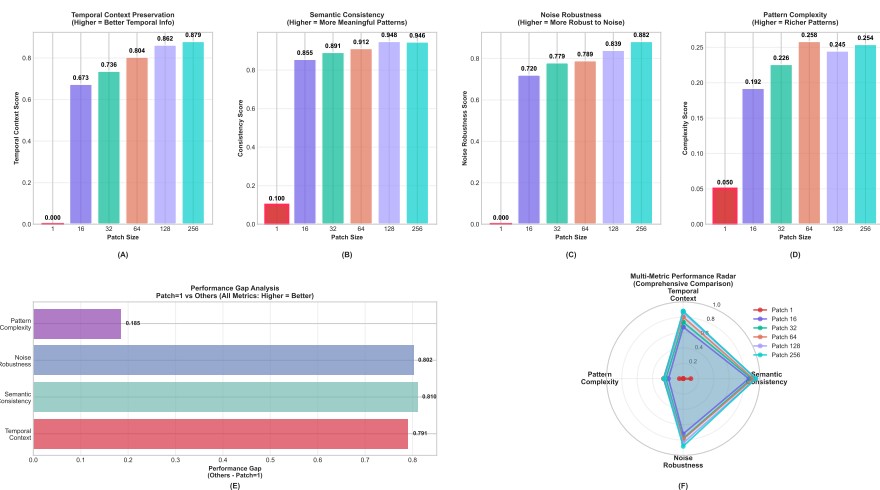

Figure 2: (A-D) show core semantic metrics: (A) Temporal Context Preservation, (B) Semantic Consistency, (C) Noise Robustness, (D) Pattern Complexity. (E) Performance gap analysis between patch=1 and larger patch sizes. (F) Multi-metric radar comparison.

**Temporal Context Preservation.** Point-based methods necessitate complex auxiliary mechanisms to reconstruct temporal ordering from isolated samples, often resulting in incomplete dependency capture. Patch-based methods naturally embed temporal structure within each segment, providing immediate access to local dynamics without additional processing overhead. This fundamental advantage makes patch-based approaches more suitable for applications requiring native temporal understanding.

**Semantic Consistency.** Point-based approaches frequently disrupt coherent temporal patterns across multiple tokens, leading to fragmented semantic interpretations. Patch-based approaches maintain semantic integrity by preserving complete local motifs as unified entities, enabling more stable and interpretable representations. This inherent consistency is particularly valuable for downstream tasks that rely on pattern recognition.

**Robustness.** Point-based representations demonstrate inherent fragility when exposed to high-frequency perturbations, stemming from their fundamental processing unit design. Patch-based representations provide intrinsic robustness through natural sample aggregation, achieving superior noise suppression while maintaining structural integrity. This inherent stability proves particularly valuable in practical deployment scenarios characterized by imperfect data quality.

**Pattern Complexity.** Point-based methods require extensive token sequences to represent complex temporal relationships, often resulting in computationally expensive processing. Patch-based methods efficiently encode multi-dimensional temporal structures within single units, providing both representational compactness and structural richness. This efficiency advantage becomes increasingly significant as pattern complexity grows.

The comparative visual analyses (*e.g.*, Panels E–F) demonstrate notable differences between patch-based and point-based approaches, with patch-based methods exhibiting more coherent radar profiles and stronger retention of temporal structure.

These empirical results provide support for our theoretical framework, indicating that patch-based approaches generally outperform point-based methods across key dimensions including temporal context preservation, semantic consistency, robustness, and pattern complexity. This performance differential suggests inherent advantages in patch-based representations for capturing temporal structure, aligning with previous experimental findings Nie et al. (2022); Chen et al. (2025).

## 2.2 THE VOCABULARY OF TIME SERIES

### 2.2.1 VOCABULARY CONSTRUCTION

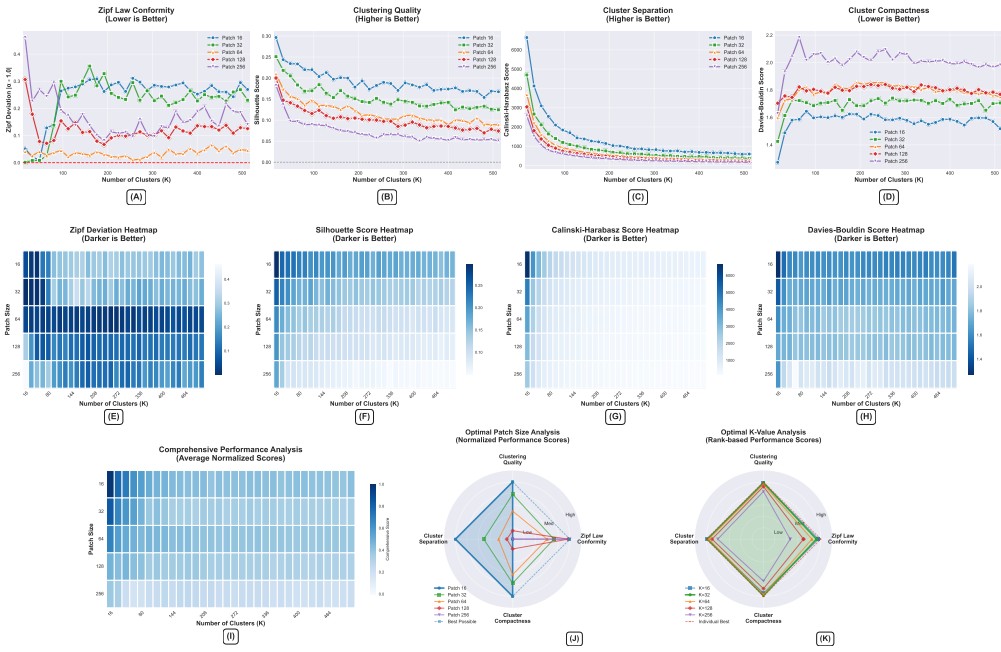

Figure 3: Performance analysis reveals trade-offs in time series vocabulary design governed by patch size ($P$) and cluster count ($K$). Panels (**A,E**) assess Zipf's law conformity, (**B,F**) evaluate clustering quality via silhouette scores, (**C,G**) analyze cluster separation, (**D,H**) measure compactness, (**I**) shows normalized performance across metrics, and (**J,K**) illustrate multi-objective trade-offs.

To empirically validate our claims, we transform continuous, high-dimensional time series into discrete, channel-agnostic representations by constructing a "Vocabulary of Time Series" composed of learnable "tokens" or "words." This data-driven vocabulary automatically discovers fundamental "Temporal Motifs"—reusable patterns that compose complex dynamic phenomena—rather than relying on predefined functions or handcrafted rules, through analysis of large-scale, heterogeneous datasets comprising approximately 1.4M patches Mueen (2014); Lonardi & Patel (2002).

Our pipeline employs a three-stage process to transform time series into discrete tokens:

1. **Patching**: Segment raw time series into continuous patches of length $P$ with stride $S$, serving as fundamental analysis units Nie et al. (2022).

2. **Quantization**: Use K-Means clustering to create a vocabulary $\mathcal{C}$ of $K$ centroids ("Temporal Words"), offering both intuitive pattern discovery and computational efficiency via Mini-Batch processing.

3. **Tokenization**: Map patches to token IDs through nearest centroid assignment, compressing data while enabling subsequent linguistic analysis of temporal patterns.

Following our experimental setup, we conducted a comprehensive analysis to evaluate the properties of the generated vocabularies. The results, summarized in Figure 3, reveal critical insights into the relationship between tokenization parameters and vocabulary quality. Our analysis of these results reveals a fundamental trade-off between the structural quality of the learned vocabulary and its statistical resemblance to natural language.

The experimental results reveal a fundamental trade-off between structural and linguistic properties:

**Structural Fidelity**  . Smaller patches consistently outperform in clustering quality metrics. "Patch 16" achieves the highest Silhouette Score and optimal Davies-Bouldin Score, with performance monotonically decreasing as patch size increases to 256. This superiority stems from the inherent characteristics of small patches: they represent simple, "atomic" patterns with low dimensionality and minimal variation, enabling K-Means to partition them into well-defined, compact, and clearly separated clusters.

**Linguistic Plausibility**  . Larger patches demonstrate superior conformity to Zipf's Law, with $P = 128$ and $P = 256$ achieving the lowest deviation from ideal Zipfian distributions. The optimal vocabulary size for linguistic plausibility ($K = 32$ or $K = 48$) is consistently higher than for structural optimization. This advantage arises because larger patches capture more complete, semantically rich "temporal motifs," creating diverse vocabularies that mirror natural language's characteristic long-tail distribution where specific low-frequency words dominate.

Our analysis reveals a trade-off in vocabulary construction: smaller patches ($P = 16, 32$) achieve superior clustering quality metrics, while larger patches ($P = 128, 256$) demonstrate better Zipf conformity. This establishes that patch size selection represents a core design decision rather than simple optimization—dictating whether we obtain simple "atomic" patterns or complex, semantically rich "motifs" with language-like statistical properties. Cluster compactness analysis further supports our "distributional token" hypothesis, as Davies-Bouldin Scores remain suboptimal across configurations, indicating clusters are inherently diffuse rather than tightly concentrated. This reflects an intrinsic data property: temporal motif boundaries are not sharply defined, with patches often exhibiting multiple characteristics simultaneously (e.g., primary trends with secondary volatilities), causing vector representations to lie in overlapping boundary regions where clusters represent probabilistic distributions rather than discrete points. Consequently, we proceed with $P = 32$ and $K = 32$ by default, recognizing this choice fundamentally shapes the vocabulary nature and subsequent learning paradigm for foundation models, given that time series lack the fixed vocabulary standard of language models.

### 2.2.2 VOCABULARY STATISTICS

Figure 4 provides a comprehensive statistical analysis of the temporal vocabulary generated by applying K-Means clustering with parameters set to a patch size of $P = 32$ and a vocabulary size of $K = 32$. The results offer compelling, multi-faceted evidence for our central hypothesis: that tokenized time series data exhibits a robust, language-like structure.

**Cluster Structure and Separability.**   The t-SNE visualization (Panel A) reveals the geometric distribution of patch embeddings from 32-dimensional space to 2D, using 50,000 sampled data points. The 32 clusters form visually coherent and partially separable groups with varying spatial organization. This demonstrates successful identification of meaningful, recurring patterns that are generally separable in high-dimensional space. The inter-cluster distance distribution (Panel D) shows mean pairwise distance of 1.82 units with median around 1.65 units, indicating moderate separation. Meanwhile, significant spatial overlap between clusters supports our "distributional token" hypothesis, suggesting temporal motifs are continuous regions rather than discrete points. The WCSS analysis (Panel E) reveals substantial variance in cluster compactness (4-10K for compact clusters, 20-30K for dispersed clusters), reflecting natural heterogeneity in temporal pattern complexity.

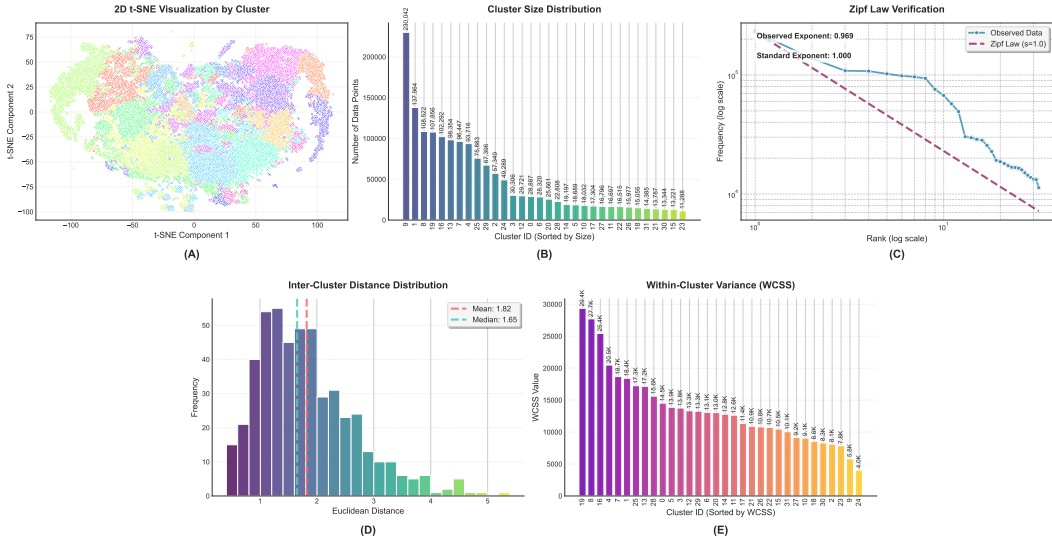

Figure 4: **Comprehensive Statistical Analysis of the Temporal Vocabulary** ($P = 32$, $K = 32$). **(A)** *Cluster Structure Visualization*: t-SNE visualization of clustered patches, where each patch is color-coded by its assigned cluster membership. **(B)** *Token Frequency Distribution*: Token frequency distribution showing the number of patches assigned to each cluster. **(C)** *Zipf Conformity Analysis*: Zipf conformity analysis comparing our temporal clustering distribution against the standard Zipf distribution. **(D)** *Inter-Cluster Distance*: Inter-cluster distance distribution measuring the separation degree between different cluster centroids. **(E)** *Within-Cluster Variance*: Within-cluster variance (WCSS) bar chart illustrating the compactness of each cluster.

**Zipfian Frequency Distribution.** The most striking finding is the vocabulary's adherence to Zipf's Law. The cluster size distribution (Panel B) shows a significant long-tail characteristic, with the largest cluster containing 230k instances and the smallest only 11.3k instances, resulting in an imbalance ratio of 20.4:1. In the log-log rank-frequency plot (Panel C), the empirical data (blue line) aligns remarkably well with the ideal Zipfian distribution (red dashed line), yielding a Zipf exponent of 0.969 with $R^2 = 0.910$. This strong power-law signature provides powerful empirical validation that complex temporal dynamics are composed from a vocabulary of reusable motifs governed by language-like statistical principles. The high-frequency patterns cover 29.6% of the data with the top 9.4% of clusters, while the long tail contributes 16.7% unique patterns from the bottom 50%, confirming both efficiency and expressive completeness.

Figure 4 establishes an empirical foundation for viewing time series through a linguistic lens. We discover that patches statistically resemble tokens in foundation models, but with key differences: (1) Time series tokens represent motif families rather than individual points, as WCSS values show clusters remain distributions; (2) Time series token distributions are inherently separable, per Inter-Cluster Distance analysis. Therefore, time series tokens constitute a distributional generalization of language model tokens. The robust Zipf-like structure, combined with distributional evidence, provides fundamental justification for applying language model paradigms to time series.

## 2.3 THE QUASI-LINGUISTIC PROPERTIES OF TIME SERIES

**Zipfian Distribution in the Temporal Lexicon** . Our experimental results strongly support this theory: across all vocabulary sizes ($K = 16$ to $256$), the frequency distribution of "temporal words" robustly follows Zipf's Law. Notably, regardless of the number of clusters (vocabulary size) chosen, the statistical patterns consistently obey Zipf's Law, aligning with the statistical regularities of natural language. As shown in Figure 5(A), the relationship between word frequency and rank on a log-log scale exhibits a clear linear pattern with exponent $\alpha \approx 1.0$, indicating universal power-law behavior across different granularities. This Zipfian characteristic is not merely statistical—it implies *compositionality* and *evolvability*, suggesting that temporal dynamics are composed from

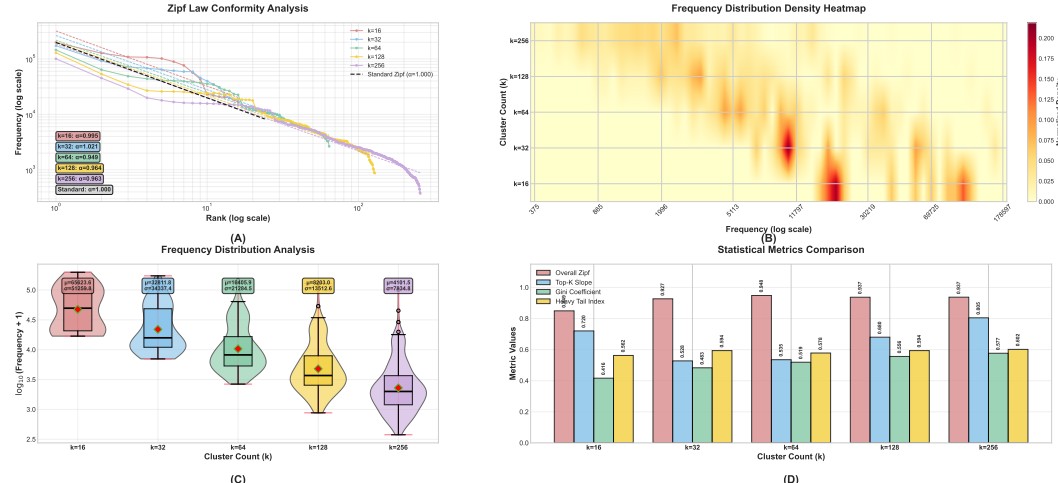

Figure 5: **Statistical analysis of quasi-linguistic properties in patch-based time series token distributions.** **(A)** *Zipf law conformity*: Log–log rank–frequency plots for $k \in \{16, 32, 64, 128, 256\}$, with fitted exponents $\alpha \approx 1$. **(B)** *Frequency density heatmap*: Normalized density of token frequencies across cluster sizes. **(C)** *Distributional statistics*: log-transformed token frequencies; larger vocabularies yield lower mean frequencies and longer tails. **(D)** *Metric-based comparison*: Quantitative evaluation of Zipf fit, top-$K$ slope, Gini coefficient, and heavy-tail index.

a finite set of reusable "primitives" through a latent "grammar," rather than being independent numerical sequences. Two key insights emerge: (1) this provides empirical grounding for treating time series analysis as language modeling, explaining Transformer effectiveness in natural language through structural isomorphism; (2) the distribution's **head** encodes universal dynamics (stability, trends, seasonality), while its **long tail** contains rare, domain-specific critical events. Foundation models must master both the universal grammar and rare knowledge.

**Robustness and Dynamic Adaptability of Vocabulary Structure** . While Zipf's Law establishes quasi-linguistic structure, we examine its stability across vocabulary sizes $K$. Figure 5(B) shows that smaller $K$ values (16, 32) produce concentrated frequency distributions with clear density peaks in the medium-to-high frequency range ($\sim 10^3$-$10^4$), while larger $K$ values (64-256) yield more dispersed distributions with increased low-frequency clusters. Figure 5(C) quantitatively confirms this pattern: $K = 16$ and $K = 32$ maintain relatively stable, concentrated distributions, while larger $K$ values exhibit greater dispersion and pronounced long tails. Crucially, the distribution's *shape* remains invariant: high-frequency "superstar" primitives persist as stable outliers across all $K$ values, confirming they represent fundamental dynamic patterns rather than statistical artifacts. This structural invariance demonstrates that our vocabulary captures genuine data properties, not parameter-dependent artifacts.

**Optimal Vocabulary Size Selection and Quantitative Validation** . Visual patterns require quantitative validation. Figure 5(D) presents four metrics: overall Zipf fit, top-$K$ slope, Gini coefficient, and heavy-tail index. All metrics show remarkable stability across $K$ values, confirming structural robustness. Overall Zipf fit remains high ($\approx 0.86$–$0.92$), with $K = 32$ achieving optimal fit (0.927). Top-$K$ slope stays near standard values, with $K = 256$ performing best on high-frequency patterns (0.808). Gini coefficient ($\approx 0.6$) indicates structured inequality ($K = 16$: 0.552), distinguishing it from random noise. Heavy-tail index shows higher values for $K = 16$ and $K = 256$ ($\approx 0.576$), quantifying the prevalence of rare patterns. This analysis reveals that smaller $K$ values (16-32) better capture dominant patterns, while larger $K$ values extract finer details. $K = 32$ emerges as the optimal balance point, achieving both the highest Zipf fit and robust overall performance. The joint metric stability elevates quasi-linguistic structure from analogy to statistical law, proving its status as an intrinsic property of temporal data. This validates the "language of time" hypothesis and recommends $K = 32$ as the optimal vocabulary size for foundation models.

We observe state inertia, sparse higher-order n-grams, and long-tailed transition probabilities. Temporal motifs are hierarchically organized, with frequent stable states anchoring the grammar while rare events act as exceptions. Transitions flow toward equilibrium, and sequences are predominantly stationary, with instability appearing only as rare, marked structures. Detailed see Appendix C.

## 3 THEORETICAL FOUNDATION

Building upon our empirical findings, we develop a comprehensive theoretical framework that addresses the fundamental question: *The theoretical guarantees of time series foundation models.*

### 3.1 FAITHFUL DISCRETIZATION

**Core Question:** Can we reliably convert continuous time series patches into discrete tokens without losing essential information?

**Theorem 3.1** ($\varepsilon$-Covering Guarantees Bounded Information Loss). *Let $0 < \varepsilon < 2\sqrt{P}$, where $P$ is the patch dimension. There exists a codebook $\mathcal{C}$ with a finite size $K$ such that for any patch vector $h$, its quantized representation $Q_{\mathcal{C}}(h)$ satisfies $d(h, Q_{\mathcal{C}}(h)) \leq \varepsilon$.*

**Intuitive Meaning:** Think of this like creating a "dictionary" for time series patterns. Just as we can represent any color using a finite palette with small error, we can represent any time series pattern using a finite vocabulary of "typical patterns" with bounded distortion. This theorem guarantees that tokenization is mathematically sound—we're not throwing away critical information.

**Supporting Theory:** This foundation is further supported by the deterministic quantization lemma (A.1), Voronoi partitioning lemma (A.2), and the detailed $\varepsilon$-covering theorem (A.1) in Appendix A.

### 3.2 NATURAL STATISTICAL STRUCTURE

**Core Question:** Why do patch tokens follow power-law (Zipf-like) distributions, and what does this tell us about time series structure?

**Theorem 3.2** (Zipf-like Long-Tail Distribution for Patch Tokens). *Assume the probability distribution of tokens follows a two-parameter GEM distribution. The expected value of its ranked empirical frequency $f_n(r)$ (the frequency of the $r$-th most common token) satisfies a power-law relationship: $\mathbb{E}[f_n(r)] \asymp r^{-\beta}$.*

**Intuitive Meaning:** Time series patterns follow a "natural hierarchy"—some patterns (like smooth trends) are very common, while complex oscillations are rare. This is exactly like natural language, where common words like "the" appear frequently while specialized terms are rare. This structure is crucial because it means we can compress most time series efficiently using a compact vocabulary.

**Supporting Theory:** This foundation is extended by the comprehensive Zipf distribution theorem (A.2) and the centroid offset analysis (A.4) in Appendix A, which provide detailed asymptotic analysis and finite-sample guarantees.

### 3.3 ENHANCED MODEL CAPACITY

**Core Question:** Do patch-based models have greater expressive power than point-based?

**Theorem 3.3** (Capacity Measure Monotonicity). *The hypothesis space of a patch-based model, $\mathcal{H}_{patch}$, contains the hypothesis space of an equivalent pointwise model, $\mathcal{H}_{point}$ (i.e., $\mathcal{H}_{point} \subseteq \mathcal{H}_{patch}$). Consequently, any standard measure of model capacity (e.g., VC Dimension, Rademacher Complexity) for the patch-based model is greater than or equal to that of the pointwise model.*

**Intuitive Meaning:** Patch models can do everything pointwise models can do, plus more. Imagine the difference between reading a book word-by-word versus phrase-by-phrase—the phrase reader can understand context and meaning that the word-by-word reader misses. Similarly, patches capture temporal patterns that individual points cannot.

**Supporting Theory:** This capacity result is strengthened by the subclass lemma (A.3), the comprehensive monotonicity theorem (A.5), and capacity monotonicity corollary (A.2) in Appendix A.

### 3.4 TEMPORAL STRUCTURE PRESERVATION

**Core Question:** Does the patching process destroy important temporal dependencies in the data?

**Lemma 3.1** ($\beta$-Mixing Preservation). *If the original time series $\{X_t\}$ is $\beta$-mixing (a common measure of temporal dependency), then the resulting token sequence $\{T_m\}$ is also $\beta$-mixing.*

**Intuitive Meaning:** The "memory structure" of time series is preserved during tokenization. If your original data has the property that distant past has diminishing influence on the present (which most well-behaved time series do), then this property is maintained after converting to tokens. This ensures we don't accidentally destroy the temporal logic of the data.

**Supporting Theory:** This preservation result is complemented by the detailed $\beta$-mixing preservation lemma (A.4) in Appendix A, which provides explicit mixing coefficient bounds.

### 3.5 GENERALIZATION GUARANTEES

**Core Question:** Can we prove that patch-based models will generalize well to unseen data?

**Theorem 3.4** (Dependence Generalisation Bound). *For a learning algorithm with uniform stability $\varepsilon_{stab}$ trained on a $\beta$-mixing token sequence of length $n$, the generalization error is bounded with high probability: $G_n(A) \leq 2\varepsilon_{stab} + O(\frac{1}{\sqrt{n}})$.*

**Intuitive Meaning:** This provides mathematical insurance that patch-based models won't overfit. The bound tells us that if our training algorithm is stable (small changes in training data lead to small changes in the model), then generalization error decreases predictably as we get more data. This is why patch methods work reliably across different domains.

**Supporting Theory:** This generalization theory is expanded by the comprehensive dependence-aware generalization theorem (A.6), the MSE predictor corollary (A.1), and the ERM risk non-increase corollary (A.3) in Appendix A.

### 3.6 INFORMATION-THEORETIC OPTIMALITY

**Core Question:** Is patching just a valid representation, or is it actually *better* than alternatives?

**Theorem 3.5** (Patch Representation as an Effective Information Bottleneck). *Patching and quantization transform the input $X$ into a compressed representation $Z_{patch}$. This process acts as an information bottleneck that preferentially discards noise (reducing the compression cost $I(X; Z_{patch})$) while preserving task-relevant information (maintaining the predictive power $I(Y; Z_{patch})$).*

**Intuitive Meaning:** Think of this like smart data compression. Regular compression (like ZIP files) removes data blindly. But patching is like an intelligent editor who removes noise and typos while keeping the meaningful content. The mathematical framework proves that patching preferentially discards irrelevant variations (noise) while preserving the patterns needed for accurate prediction.

**Supporting Theory:** This optimality result is deepened by the comprehensive information bottleneck theorem (A.7), the mutual information preservation theorem (A.8), and the information bottleneck sufficiency corollary (A.4) in Appendix A.

## 4 CONCLUSION

In this paper, we began by addressing the paradox of why time series foundation models transfer so well across different domains and proposed an analysis of these models from a language model perspective. Through rigorous empirical and theoretical analysis, we conclude that patch-based time series foundation models are essentially a variant of a language model with distributional tokens. This helps these models inherit the advantages of language models, enabling them to have better transfer and representation abilities. Additionally, we found several interesting features of time series, which are referred to as "the grammar of time series." Overall, this paper provides a theoretical bound that affirms the potential of time series foundation models and encourages future researchers to analyze time series from a structured perspective, rather than solely from a numerical one.

ETHICS STATEMENT

This study only uses publicly available and de-identified time series data. It does not involve personal privacy or human subject experiments, and there are no conflicts of interest. Therefore, no additional ethical approval is required.

REPRODUCIBILITY STATEMENT

The paper provides a complete list of datasets, preprocessing methods, tokenization and clustering procedures, as well as experimental metrics and hyperparameter settings. The appendix contains all necessary formulas and implementation details sufficient to reproduce the experimental results. We provide the source code at: https://anonymous.4open.science/r/The-language-of-time-750A/.

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

## THE USE OF LLM

In this study, large language models (LLMs) were used to polish the writing of the paper and to assist in the implementation of part of the code.

## A   HIERARCHICAL MATHEMATICAL PROOF FRAMEWORK: KEY CONTRIBUTIONS AND RESULTS

Table 2: Summary of the Theoretical Chain Supporting Patch Quantization

| Objective | Key Result | Core Conclusion | Interpretation |
|---|---|---|---|
| Faithful discretization | **Thm. 3.1** ($\varepsilon$-Covering) | A finite codebook exists, guaranteeing quantization error for any patch is bounded by $\leq \varepsilon$. | Time series patterns can be reliably discretized without losing essential information. |
| Natural statistical structure | **Thm. 3.2** (Zipf-like distribution) | Patch tokens follow a power-law rank-frequency relationship: $\mathbb{E}[f_n(r)] \asymp r^{-\beta}$. | Time series patches exhibit natural language-like statistical hierarchy. |
| Enhanced model capacity | **Thm. 3.3** (Capacity Monotonicity) | Patch-based models contain pointwise models as a subset: $\mathcal{H}_{\text{point}} \subseteq \mathcal{H}_{\text{patch}}$. | Patches can express everything points can, plus temporal patterns. |
| Cluster-plus-offset mechanism | **Thm. A.3 & A.4** (Classification-Regression Decomposition) | Patch prediction decomposes into classification (select cluster) plus regression (predict offset): $P(\mathbf{x}_{t+1} \mid \mathbf{x}_{\leq t}) = \sum_k P(K = k \mid \mathbf{x}_{\leq t}) \cdot P(\boldsymbol{\epsilon}_k \mid K = k, \mathbf{x}_{\leq t})$ | Complex temporal forecasting reduces to discrete pattern recognition plus continuous fine-tuning. |
| Temporal structure preservation | **Lem. 3.1** ($\beta$-Mixing Preservation) | If the original time series is $\beta$-mixing, then token sequence remains $\beta$-mixing. | Patching preserves the memory structure of time series data. |
| Generalization guarantees | **Thm. 3.4** (Dependence Gen. Bound) | For stable algorithms on $\beta$-mixing sequences: $G_n(A) \leq 2\varepsilon_{\text{stab}} + O(\frac{1}{\sqrt{n}})$. | Patch models generalize predictably without overfitting. |
| Information-theoretic optimality | **Thm. 3.5** (Information Bottleneck) | Patching acts as intelligent compression that discards noise while preserving task-relevant information. | Patches provide optimal balance between compression and prediction. |

### A.1   THEORETICAL ANALYSIS OF CAPABILITY

#### A.1.1   $\varepsilon$-STATISTICAL SUFFICIENCY

**Theorem A.1** ($\varepsilon$-Covering Guarantees Bounded Information Loss). *Let $d(h, c) = \|h - c\|_2$ be the Euclidean (chordal) metric on the centered subspace and let $0 < \varepsilon \leq 2\sqrt{P}$. There exists a prototype set (codebook) $\mathcal{C} \subset \mathcal{H}^P$ with size $K$ bounded by*

$$K \ \leq \ \left(1 + \frac{2\sqrt{P}}{\varepsilon}\right)^{P-1}, \tag{1}$$

*such that for any patch vector $h \in \mathcal{H}^P$,*

$$d\big(h, Q_{\mathcal{C}}(h)\big) \ = \ \min_{c \in \mathcal{C}} \|h - c\|_2 \ \leq \ \varepsilon. \tag{2}$$

*Hence the discrete token $T = Q_{\mathcal{C}}(H)$ is an $\varepsilon$-**faithful representation under $L_2$ distortion**: the worst-case input distortion is bounded by $\varepsilon$.*

*Proof.* **Embedding into the Unit Sphere.** We assume the patch space $\mathcal{H}^P$ consists of centered, norm-normalized vectors $h \in \mathbb{R}^P$ with $\sum_i h_i = 0$ and $\|h\|_2 = \sqrt{P}$. Thus $\mathcal{H}^P$ lies on the radius-$\sqrt{P}$ sphere in a $(P-1)$-dimensional subspace (isometric to $S^{P-2}$). The scaling $h \mapsto h' = h/\sqrt{P}$ is an isometry up to factor $1/\sqrt{P}$: for all $h_1, h_2$,

$$\|h_1' - h_2'\|_2 \;=\; \frac{1}{\sqrt{P}}\,\|h_1 - h_2\|_2. \tag{3}$$

Therefore an $\varepsilon$-cover in the original space is equivalent to an $\varepsilon' = \varepsilon/\sqrt{P}$-cover on the unit sphere.

**Covering Number of the Sphere.** A standard volumetric bound yields that the $\varepsilon'$-covering number of the unit sphere $S^{d-1}$ in $\mathbb{R}^d$ under $\|\cdot\|_2$ satisfies

$$N(\varepsilon', S^{d-1}) \;\leq\; \left(1 + \tfrac{2}{\varepsilon'}\right)^d. \tag{4}$$

Here $d = P - 1$, hence by substituting into Equation equation 4, we obtain

$$K \;=\; N\!\left(\varepsilon/\sqrt{P},\; S^{P-2}\right) \;\leq\; \left(1 + \frac{2\sqrt{P}}{\varepsilon}\right)^{P-1}. \tag{5}$$

This guarantees the existence of a codebook $\mathcal{C}$ achieving the stated uniform distortion bound. $\qquad\square$
$\square$

**Remark A.1** (On the Practical Construction of the Codebook). *The argument is existential (worst-case over the entire sphere). For finite datasets, algorithms such as $k$-means++ or Lloyd–Max can produce codebooks that* empirically *meet the $\|h - Q_{\mathcal{C}}(h)\|_2 \leq \varepsilon$ target. For worst-case (uniform) guarantees over $\mathcal{H}^P$, constructive coverings (e.g., spherical codes or farthest-point $k$-center heuristics) are more appropriate.*

**Discussion and Corollaries. Finite Dictionary Size.** For small $\varepsilon$, the bound in Equation equation 5 scales as

$$K(\varepsilon) = \mathcal{O}\!\left(\left(\tfrac{\sqrt{P}}{\varepsilon}\right)^{P-1}\right) = \mathcal{O}(\varepsilon^{-(P-1)}). \tag{6}$$

**Bounded Quantization Error (MSE).** If the distortion is $L_2$ and $D = \mathbb{E}\|h - Q_{\mathcal{C}}(h)\|_2^2$, then

$$D \;\leq\; \varepsilon^2. \tag{7}$$

### A.1.2 ZIPF-LIKE LONG-TAIL FREQUENCY

**Theorem A.2** (Zipf-like Long-Tail Distribution for Patch Tokens). *Assume the probability distribution of tokens $\pi$ follows a two-parameter GEM (Griffiths-Engen-McCloskey) distribution, denoted $\pi \sim GEM(d, \theta)$, with parameters satisfying $0 \leq d < 1$ and $\theta > -d$. For an i.i.d. sequence of tokens $T_1, T_2, \cdots \sim \pi$, the expected value of the ranked probabilities $\mathbb{E}[\pi_{(r)}]$ (i.e., the expected probability of the $r$-th most common token) satisfies a power-law relationship:*

$$\mathbb{E}[\pi_{(r)}] \asymp r^{-\beta}, \tag{8}$$

*where the power-law exponent $\beta$ depends on the parameter $d$:*

- *For $0 < d < 1$, the exponent is $\beta = 1/d$.*

- *For $d = 0$ (the Dirichlet Process case), the exponent is $\beta = 2$.*

*Note: The symbol $\asymp$ denotes asymptotic equivalence, i.e., $\lim_{r \to \infty} \frac{\mathbb{E}[\pi_{(r)}]}{r^{-\beta}} = C$ for some positive constant $C$.*

*Proof.* We establish this result through the connection between the GEM distribution and the Pitman-Yor process, which provides a framework for analyzing the ranked probabilities.

**Connection to Pitman-Yor and Poisson-Dirichlet Distributions:** The probability distribution $\pi = (\pi_1, \pi_2, \dots)$ generated by the $\text{GEM}(d, \theta)$ process is equivalent to the distribution of weights in a Pitman-Yor process, denoted $\text{PY}(d, \theta)$. The set of ranked probabilities $\{\pi_{(1)}, \pi_{(2)}, \dots\}$ of this process follows the two-parameter Poisson-Dirichlet distribution, $\text{PD}(d, \theta)$. The asymptotic behavior of these ranked probabilities is well-studied.

**Asymptotic Analysis for $d > 0$:** The cornerstone result for the Pitman-Yor process, established in the work of Pitman and Yor, shows that for $0 < d < 1$, the expected ranked probabilities follow a power law. For large $r$, this is given by:

$$\mathbb{E}[\pi_{(r)}] \asymp r^{-1/d}. \tag{9}$$

**Asymptotic Analysis for $d = 0$ (Dirichlet Process):** When $d = 0$, the process degenerates to the Dirichlet Process, and the ranked probabilities follow the $\mathrm{PD}(0, \theta)$ distribution. In this case, the asymptotic behavior changes. The expected ranked probabilities exhibit a different power-law decay, given by:

$$\mathbb{E}[\pi_{(r)}] \asymp r^{-2}. \tag{10}$$

This result stems from the analysis of Ewens's sampling formula, which describes the partition structure of the Dirichlet Process.

**From Theoretical Probabilities to Empirical Frequencies:** For a sequence of $n$ i.i.d. samples from the distribution $\pi$, the sequence is exchangeable. By the strong law of large numbers for exchangeable sequences, the empirical frequency of the $r$-th most frequent token, $f_n(r)$, converges to the true ranked probability $\pi_{(r)}$ almost surely as $n \to \infty$.

$$\lim_{n \to \infty} f_n(r) = \pi_{(r)}, \tag{11}$$

almost surely holds.

Therefore, for large $n$, the expectation of the empirical frequency is well-approximated by the expectation of the true probability, $\mathbb{E}[f_n(r)] \approx \mathbb{E}[\pi_{(r)}]$. This allows us to apply the asymptotic results from Equations equation 9 and equation 10 directly to $\mathbb{E}[f_n(r)]$, establishing the power-law relationships as stated in Equation equation 8. This completes the proof. Q.E.D. □

**Remark A.2** (Connection to Zipf's Law in Linguistics). *Zipf's law was first discovered in linguistics, where the power-law exponent $\beta$ is approximately 1. In our model, as the discount parameter $d \to 1^-$, we get $\beta = 1/d \to 1$, which corresponds perfectly to the classic law. Empirical studies have shown that for real-world languages, the value of $d$ is typically between $0.7$–$0.8$, which leads to $\beta \approx 1.25$–$1.4$, in high agreement with linguistic observations.*

This section's theoretical analysis provides a rigorous foundation for tokenizing continuous patch data. In short, the two lemmas establish a complete theoretical chain.

First, Theorem A.1 proves that any continuous patch can be represented by a token from a finite codebook with a guaranteed, bounded error ($\epsilon$-sufficiency). This confirms the feasibility and fidelity of the tokenization process. Second, Theorem A.2 demonstrates that if the token generation process follows a GEM distribution, the resulting token frequencies will exhibit a Zipf-like power-law distribution, a key statistical signature of natural language.

Collectively, these results provide a solid theoretical basis for treating continuous signals as a "language," thereby validating the application of powerful sequence models like the Transformer.

### A.1.3 CLASSIFICATION–REGRESSION DECOMPOSITION FOR TIME-SERIES PATCH PREDICTION

**Notation** Let the history (context) be $\mathbf{x}_{\leq t}$ and the next patch be $\mathbf{x}_{t+1} \in \mathbb{R}^P$. Assume we have $K$-means centroids (a codebook)

$$\mathcal{C} = \{\mathbf{c}_k\}_{k=1}^K, \qquad Q(\mathbf{x}) = \arg \min_{1 \leq j \leq K} \|\mathbf{x} - \mathbf{c}_j\|_2. \tag{12}$$

Define the Voronoi cells

$$V_k = \bigcap_{j \neq k} \Big\{\mathbf{x} : \|\mathbf{x} - \mathbf{c}_k\|_2 \leq \|\mathbf{x} - \mathbf{c}_j\|_2\Big\} = \bigcap_{j \neq k} \Big\{\mathbf{x} : 2(\mathbf{c}_j - \mathbf{c}_k)^\top \mathbf{x} \leq \|\mathbf{c}_j\|_2^2 - \|\mathbf{c}_k\|_2^2\Big\}. \tag{13}$$

Ties are resolved by a deterministic tie-breaking rule so that $Q$ is a well-defined measurable map. Unless otherwise noted, equalities below hold almost everywhere (ignoring measure-zero cell boundaries). To match the notation style of this paper, we use $P(\cdot)$ for conditional distribution/density; the exact type is clear from context.

**Lemma A.1** (Deterministic cluster assignment). *For any* $\mathbf{x} \in \mathbb{R}^P$, *the quantizer* $Q(\mathbf{x}) = \arg\min_k \|\mathbf{x} - \mathbf{c}_k\|_2$ *defines a deterministic mapping*

$$Q : \mathbb{R}^P \to \{1, 2, \ldots, K\}. \tag{14}$$

**Lemma A.2** (Voronoi partition). *The family* $\{V_k\}_{k=1}^K$ *forms a measurable partition of* $\mathbb{R}^P$:

$$\mathbb{R}^P = \bigcup_{k=1}^K V_k, \qquad V_k \cap V_j = \emptyset \ (k \neq j), \tag{15}$$

*and* $Q(\mathbf{x}) = k \iff \mathbf{x} \in V_k$.

**Theorem A.3** (Full classification–regression decomposition for patch prediction). *Let the discrete latent variable be* $K := Q(\mathbf{x}_{t+1})$. *For any history* $\mathbf{x}_{\leq t}$,

$$P(\mathbf{x}_{t+1} \mid \mathbf{x}_{\leq t}) = \sum_{k=1}^K P\big(K = k \mid \mathbf{x}_{\leq t}\big) \cdot P\big(\mathbf{x}_{t+1} \mid K = k, \mathbf{x}_{\leq t}\big), \tag{16}$$

*where*

$$P\big(K = k \mid \mathbf{x}_{\leq t}\big) = P\big(\mathbf{x}_{t+1} \in V_k \mid \mathbf{x}_{\leq t}\big), \quad P\big(\mathbf{x}_{t+1} \mid K = k, \mathbf{x}_{\leq t}\big) = P\big(\mathbf{x}_{t+1} \mid \mathbf{x}_{t+1} \in V_k, \mathbf{x}_{\leq t}\big). \tag{17}$$

*Proof.* By the law of total probability with $K = Q(\mathbf{x}_{t+1})$,

$$P(\mathbf{x}_{t+1} \mid \mathbf{x}_{\leq t}) = \sum_{k=1}^K P\big(\mathbf{x}_{t+1}, K = k \mid \mathbf{x}_{\leq t}\big) = \sum_{k=1}^K P\big(\mathbf{x}_{t+1} \mid K = k, \mathbf{x}_{\leq t}\big) P\big(K = k \mid \mathbf{x}_{\leq t}\big). \tag{18}$$

By Lemma A.2, the event $\{K = k\}$ is equivalent to $\{\mathbf{x}_{t+1} \in V_k\}$; substituting Equation equation 18 into the definitions in Equation equation 17 yields Equation equation 16. $\square$ $\square$

**Theorem A.4** (Centroid–offset decomposition). *Given* $K = k$ *and the history* $\mathbf{x}_{\leq t}$, *define the offset*

$$\boldsymbol{\epsilon}_k := \mathbf{x}_{t+1} - \mathbf{c}_k. \tag{19}$$

*Then (translation change of variables)*

$$P\big(\mathbf{x}_{t+1} \mid K = k, \mathbf{x}_{\leq t}\big) = P_{\boldsymbol{\epsilon}_k}\big(\mathbf{x}_{t+1} - \mathbf{c}_k \mid K = k, \mathbf{x}_{\leq t}\big), \tag{20}$$

*and combining with Theorem A.3 gives the three-level decomposition*

$$P(\mathbf{x}_{t+1} \mid \mathbf{x}_{\leq t}) = \sum_{k=1}^K \underbrace{P\big(K = k \mid \mathbf{x}_{\leq t}\big)}_{\text{Classification: select a motif}} \cdot \underbrace{P_{\boldsymbol{\epsilon}_k}\big(\mathbf{x}_{t+1} - \mathbf{c}_k \mid K = k, \mathbf{x}_{\leq t}\big)}_{\text{Regression: predict the offset}}. \tag{21}$$

*Proof.* For fixed $k$, the map $\mathbf{x} \mapsto \boldsymbol{\epsilon}_k = \mathbf{x} - \mathbf{c}_k$ is bijective with unit Jacobian; conditional distributions are preserved under translation, yielding equation 20. Plugging into equation 16 gives equation 21. $\square$ $\square$

**Corollary A.1** (Bayes-optimal prediction under MSE). *Under mean squared error, the Bayes predictor is the conditional mean:*

$$\mathbb{E}[\mathbf{x}_{t+1} \mid \mathbf{x}_{\leq t}] = \sum_{k=1}^K P\big(K = k \mid \mathbf{x}_{\leq t}\big) \Big(\mathbf{c}_k + \mathbb{E}[\boldsymbol{\epsilon}_k \mid K = k, \mathbf{x}_{\leq t}]\Big). \tag{22}$$

**Remark A.3** (Soft vs. hard decisions and implementation). *Equation equation 21 corresponds to a "soft" mixture prediction; a "hard" decision* $\hat{k} = \arg\max_k P(K = k \mid \mathbf{x}_{\leq t})$ *yields the point estimate*

$$\hat{\mathbf{x}}_{t+1} = \mathbf{c}_{\hat{k}} + \widehat{\boldsymbol{\epsilon}}_{\hat{k}}(\mathbf{x}_{\leq t}), \tag{23}$$

*where* $\widehat{\boldsymbol{\epsilon}}_{\hat{k}}$ *is produced by an in-cluster regression head. Soft decisions typically mitigate error amplification from misclassification near Voronoi boundaries.*

## A.2 Non-decreased Representational Capacity and Non-increased Optimal-ERM Risk

### A.2.1 Monotone Representational Capacity

**Definition A.1** (Pointwise Hypothesis Space). *Let $X \subseteq \mathbb{R}^d$ be the input space and $Y$ be the output space. The pointwise hypothesis space is defined as:*

$$\mathcal{H}_{point} = \{h_\theta : X \to Y \mid h_\theta(x) = g_\theta(x), \theta \in \Theta\} \tag{24}$$

*where $g_\theta : \mathbb{R}^d \to Y$ is a parameterized function family as defined in Equation equation 24.*

**Definition A.2** (Patch Hypothesis Space). *Given a dictionary $\mathcal{C} = \{c_1, c_2, \ldots, c_K\}$ where each $c_i$ is a patch of length $P$, and a sliding window stride $S$, we define:*

- *Quantization function: $Q_\mathcal{C} : \mathbb{R}^d \to \mathcal{C}^*$, which segments the input sequence and maps it to the nearest patches in the dictionary*

- *Embedding function: $e : \mathcal{C} \to \mathbb{R}^{d'}$, which maps patch tokens to the embedding space*

- *Reconstruction function: $Reconstruct : (\mathbb{R}^{d'})^* \to \mathbb{R}^d$, which reconstructs patch sequences to the original dimension*

*The patch hypothesis space is defined as:*

$$\begin{aligned} \mathcal{H}_{patch} = \Big\{ h_{\theta,\mathcal{C}}^{patch} : X \to Y \,\Big|\, & h_{\theta,\mathcal{C}}^{patch}(x) \\ & = g_\theta\big(Reconstruct(Embed(Q_\mathcal{C}(x)))\big), \quad \theta \in \Theta \Big\}. \end{aligned} \tag{25}$$

**Assumption A.1.** *We assume the following conditions hold:*

1. *$g_\theta$ is a continuous function for all $\theta \in \Theta$*

2. *There exists an inverse reconstruction function such that under specific conditions, $Reconstruct(Embed(\cdot))$ can be the identity mapping*

3. *The parameter space $\Theta$ remains consistent across both hypothesis spaces*

**Lemma A.3** (Idealized Subclass Relation). *Let the dictionary $\mathcal{C}$ contain all length-1 patches, i.e., $\mathcal{C} \supseteq \{(x_i) \mid x_i \in \mathbb{R}, i = 1, \ldots, d\}$, and set the sliding window stride $S = 1$. Under appropriate choices of embedding and reconstruction functions, there exists:*

$$\mathcal{H}_{point} \subseteq \mathcal{H}_{patch} \tag{26}$$

*Proof.* **Construct the special embedding function.** For any length-1 patch $c = (a) \in \mathbb{R}$, define the embedding function as:

$$e(c) = a \in \mathbb{R} \tag{27}$$

i.e., the identity embedding.

**Verify the identity property of reconstruction.** When $S = 1$ and all length-1 patches are in the dictionary, for any $x = (x_1, x_2, \ldots, x_d) \in \mathbb{R}^d$:

- Quantization process: $Q_\mathcal{C}(x) = ((x_1), (x_2), \ldots, (x_d))$

- Embedding process: $Embed(Q_\mathcal{C}(x)) = (e(x_1), e(x_2), \ldots, e(x_d)) = (x_1, x_2, \ldots, x_d)$

- Reconstruction process: $Reconstruct(Embed(Q_\mathcal{C}(x))) = x$

**Establish functional equivalence.** For any $h_\theta^{\text{point}} \in \mathcal{H}_{\text{point}}$, there exists a corresponding $h_{\theta,\mathcal{C}}^{\text{patch}} \in \mathcal{H}_{\text{patch}}$ such that:

$$h_{\theta,\mathcal{C}}^{\text{patch}}(x) = g_\theta(Reconstruct(Embed(Q_\mathcal{C}(x)))) = g_\theta(x) = h_\theta^{\text{point}}(x) \tag{28}$$

Therefore, Equation equation 26 holds. □ □

**Theorem A.5** (Capacity Measure Monotonicity). *Let $\mathcal{H}_1 \subseteq \mathcal{H}_2$ be two hypothesis spaces. Then:*

1. ***VC Dimension Monotonicity****: $\mathrm{VC}(\mathcal{H}_1) \leq \mathrm{VC}(\mathcal{H}_2)$*

2. ***Empirical Rademacher Complexity Monotonicity****: $\widehat{\mathfrak{R}}_n(\mathcal{H}_1) \leq \widehat{\mathfrak{R}}_n(\mathcal{H}_2)$*

*Proof.* **VC Dimension:** Let $\mathcal{S} = \{x_1, \ldots, x_m\}$ be any set shattered by $\mathcal{H}_1$. That is, there exist functions $h_1, \ldots, h_{2^m} \in \mathcal{H}_1$ such that for each $A \subseteq \{1, \ldots, m\}$, there exists $h_A \in \mathcal{H}_1$ satisfying:

$$h_A(x_i) = \begin{cases} 1 & \text{if } i \in A \\ 0 & \text{if } i \notin A \end{cases} \tag{29}$$

Since $\mathcal{H}_1 \subseteq \mathcal{H}_2$, all these functions also belong to $\mathcal{H}_2$, hence $\mathcal{S}$ is also shattered by $\mathcal{H}_2$. Therefore, $\mathrm{VC}(\mathcal{H}_1) \leq \mathrm{VC}(\mathcal{H}_2)$.

**Rademacher Complexity Part:**

$$\begin{aligned} \widehat{\mathfrak{R}}_n(\mathcal{H}_1) &= \mathbb{E}_\sigma \left[ \sup_{h \in \mathcal{H}_1} \frac{1}{n} \sum_{i=1}^{n} \sigma_i h(x_i) \right] \\ &\leq \mathbb{E}_\sigma \left[ \sup_{h \in \mathcal{H}_2} \frac{1}{n} \sum_{i=1}^{n} \sigma_i h(x_i) \right] \\ &= \widehat{\mathfrak{R}}_n(\mathcal{H}_2) \end{aligned} \tag{30}$$

where the inequality follows from $\mathcal{H}_1 \subseteq \mathcal{H}_2$, making the supremum over $\mathcal{H}_2$ at least as large as that over $\mathcal{H}_1$, as shown in Equation equation 30. $\square$ $\square$

**Corollary A.2** (Capacity Monotonicity for Patch Methods). *Combining Lemma A.3 and Theorem A.5, under the stated conditions:*

1. $\mathrm{VC}(\mathcal{H}_{patch}) \geq \mathrm{VC}(\mathcal{H}_{point})$

2. $\widehat{\mathfrak{R}}_n(\mathcal{H}_{patch}) \geq \widehat{\mathfrak{R}}_n(\mathcal{H}_{point})$

**Remark A.4** (Critical Limitation: Infinite Dictionary Requirement). *The assumption that $\mathcal{C}$ contains all length-1 patches from $\mathbb{R}$ requires an uncountably infinite dictionary, making this theoretical result impractical. In reality:*

- *Any practical dictionary must be finite*

- *The theoretical guarantee only holds approximately with quantization error*

- *The result should be interpreted as an idealized upper bound rather than a practical guarantee*

**Remark A.5** (Generalization Bounds). *While patch methods possess higher representational capacity, this may lead to:*

- *Larger generalization error upper bounds*

- *Requirements for more training data to achieve comparable generalization performance*

**Remark A.6** (Practical Trade-offs). *The theoretical capacity advantage must be balanced against:*

- *Computational efficiency*

- *Memory requirements*

- *Optimization difficulty*

**Proposition A.1** (Extension to Other Measures). *The monotonicity results above extend to:*

- ***Pseudo-dimension****: For real-valued function classes*

- ***Gaussian complexity****: Using Gaussian random variables instead of Rademacher variables*

- *Local Rademacher complexity*: *Defined over subsets of function classes*

*The proof methodology follows similarly, based on the monotonicity of set inclusion relations.*

**Remark A.7** (Connection to PatchTST). *The "P=1" ablation study in PatchTST Nie et al. (2022) corresponds exactly to the setup described in Lemma A.3, where the original sequence is treated as "minimal patches." This validates the practical relevance of our theoretical framework.*

### A.2.2 NON-INCREASED OPTIMAL-ERM RISK

**Corollary A.3** (Optimal-ERM Risk Non-Increase). *Under the inclusion $\mathcal{H}_{point} \subseteq \mathcal{H}_{patch}$ (Lemma A.3), for the same dataset $D$ and loss function $\ell$,*

$$\min_{h \in \mathcal{H}_{patch}} \widehat{R}_D(h) \le \min_{h \in \mathcal{H}_{point}} \widehat{R}_D(h). \tag{31}$$

*Proof.* **Define the optimal pointwise hypothesis.** Let

$$h^{\star}_{\text{point}} = \arg \min_{h \in \mathcal{H}_{\text{point}}} \widehat{R}_D(h). \tag{32}$$

Even if the minimum is not attained, an approximating sequence suffices.

**Lift to the patch class.** By Lemma A.3 (Equation equation 26), we have $h^{\star}_{\text{point}} \in \mathcal{H}_{\text{patch}}$.

**Compare minima over classes.** The minimum over a superset satisfies:

$$\min_{h \in \mathcal{H}_{\text{patch}}} \widehat{R}_D(h) \le \widehat{R}_D(h^{\star}_{\text{point}}) = \min_{h \in \mathcal{H}_{\text{point}}} \widehat{R}_D(h). \tag{33}$$

By Equation equation 33, the minimal empirical risk in the larger patch class is no greater than that in the smaller pointwise class, establishing Equation equation 31. □ □

**Remark A.8** (Scope and Limitations). *The inclusion in Lemma A.3 relies on an uncountable dictionary and identity reconstruction (stride $S = 1$, patch length $P = 1$). Hence Equation equation 31 should be interpreted as an* idealized upper bound *rather than a practical guarantee.*

### A.3 A RIGOROUS BOUND FOR TOKEN SEQUENCE DEPENDENCE

**Definition A.3** (Patch Construction). *Given a time series $\{X_t\}$, we define the patch sequence $\{Z_m\}$ as:*

$$Z_m = \left(X_{(m-1)S+1}, X_{(m-1)S+2}, \dots, X_{(m-1)S+P}\right) \tag{34}$$

*where $S > 0$ is the stride and $P > 0$ is the patch size.*

**Definition A.4** (Quantization). *Through a deterministic quantization function $Q : \mathbb{R}^P \to \mathcal{T}$, where $\mathcal{T}$ is the token space, we obtain the token sequence $\{T_m\}$:*

$$T_m = Q(Z_m) \tag{35}$$

**Lemma A.4** (Patch $\beta$-Mixing Preservation). *Let $\{X_t\}$ be a $\beta$-mixing sequence with coefficients satisfying $\beta_X(k) \le Ce^{-\rho k}$ for some constants $C, \rho > 0$. The token sequence $\{T_m\}$ constructed via equation 34 and equation 35 remains $\beta$-mixing. Furthermore, when the non-overlapping condition $S \ge P$ holds, its $\beta$-mixing coefficients are bounded by:*

$$\beta_T(k) \le \beta_X(kS - P + 1) \tag{36}$$

*Proof.* The proof proceeds in four steps.

$\sigma$**-Algebra Setup.** To determine the $\beta$-mixing coefficient $\beta_T(k)$ for the token sequence, we consider the $\sigma$-algebras representing the past and future of the sequence $\{T_m\}$:

$$\begin{aligned} \mathcal{F}_m &= \sigma(T_1, T_2, \dots, T_m) \\ \mathcal{G}_{m+k} &= \sigma(T_{m+k}, T_{m+k+1}, \dots) \end{aligned} \tag{37}$$

Since each token $T_j$ is a deterministic function of the patch $Z_j = (X_{(j-1)S+1}, \dots, X_{(j-1)S+P})$, these $\sigma$-algebras are contained within the $\sigma$-algebras of the original sequence $\{X_t\}$. Specifically,

the last data point influencing $\mathcal{F}_m$ is $X_{(m-1)S+P}$, and the first data point influencing $\mathcal{G}_{m+k}$ is $X_{(m+k-1)S+1}$. This gives us the tightest possible inclusions:

$$\mathcal{F}_m \subseteq \sigma(X_{-\infty}, \ldots, X_{(m-1)S+P}) \tag{38}$$

$$\mathcal{G}_{m+k} \subseteq \sigma(X_{(m+k-1)S+1}, \ldots, X_{\infty}) \tag{39}$$

**Temporal Gap Analysis.** The temporal gap between the two $\sigma$-algebras of the underlying process in equation 38 and equation 39 is the difference between the first index of the future and the last index of the past:

$$\text{Gap} = ((m+k-1)S+1) - ((m-1)S+P) = kS - P + 1 \tag{40}$$

Given the condition $S \geq P$ and $k \geq 1$, this gap is guaranteed to be positive, as $kS - P + 1 \geq S - P + 1 \geq 1$.

**$\beta$ - Mixing Inequality Derivation.** By the definition of the $\beta$-mixing coefficient for $\{T_m\}$ using the $\sigma$-algebras from equation 37, we have:

$$\beta_T(k) = \sup_m \sup_{\substack{A \in \mathcal{F}_m \\ B \in \mathcal{G}_{m+k}}} |\mathbb{P}(A \cap B) - \mathbb{P}(A)\mathbb{P}(B)| \tag{41}$$

Since $T_m$ is a deterministic function of $X_t$, any events $A \in \mathcal{F}_m$ and $B \in \mathcal{G}_{m+k}$ correspond to preimage events in the appropriate $\sigma$-algebras of $\{X_t\}$. The dependence cannot be increased by this deterministic transformation. Therefore, the dependence between $A$ and $B$ is bounded by the dependence between their preimages, separated by the gap from equation 40:

$$|\mathbb{P}(A \cap B) - \mathbb{P}(A)\mathbb{P}(B)| \leq \sup_{\substack{A' \in \sigma(X_{-\infty}, \ldots, X_{(m-1)S+P}) \\ B' \in \sigma(X_{(m+k-1)S+1}, \ldots)}} |\mathbb{P}(A' \cap B') - \mathbb{P}(A')\mathbb{P}(B')|. \tag{42}$$

The right-hand side of equation 42 is precisely the definition of the $\beta$-mixing coefficient of the original sequence $\{X_t\}$ for a gap of $kS - P + 1$. Thus, we obtain the main bound:

$$\beta_T(k) \leq \beta_X(kS - P + 1) \tag{43}$$

**Exponential Decay Preservation.** Given that $\beta_X(k) \leq Ce^{-\rho k}$, we can bound $\beta_T(k)$ using equation 43:

$$\beta_T(k) \leq \beta_X(kS - P + 1) \leq Ce^{-\rho(kS-P+1)} \tag{44}$$

We can rewrite the bound from equation 44 to show that $\{T_m\}$ also exhibits exponential decay:

$$Ce^{-\rho(kS-P+1)} = Ce^{-\rho(S-P+1)}e^{-\rho S(k-1)} = C'e^{-\rho'(k-1)} \tag{45}$$

where the new constants are $C' = Ce^{-\rho(S-P+1)}$ and $\rho' = \rho S$. This confirms that the exponential decay property is preserved. $\qquad\square$

**Remark A.9** (Non-overlapping Condition). *The condition $S \geq P$ is crucial for this clean derivation. It ensures that the patches of the original time series used to generate different tokens do not overlap and, more formally, guarantees a positive temporal gap ($kS - P + 1 \geq 1$) for all $k \geq 1$. This simplifies the temporal gap analysis significantly. This is a common setup in applications like Vision Transformers (ViT).*

**Remark A.10** (Overlapping Case). *When $S < P$, the patches overlap, and the analysis becomes more complex as dependencies from shared data points must be accounted for. A more refined analysis, beyond the scope of this proof, could yield a bound such as:*

$$\beta_T(k) \leq \max\{P - S + 1, 1\} \cdot \beta_X(\max\{(k-1)S - P + 1, 1\}) \tag{46}$$

**Remark A.11** (Quantization Independence). *This result holds for any deterministic quantization function $Q$. The specific choice of tokenizer or quantization method (e.g., k-means clustering, VQ-VAE) does not affect the validity of the bound, making it broadly applicable.*

**Theorem A.6** (Dependence Generalisation Bound). *Let an algorithm $A$ have uniform stability $\varepsilon_{stab}$. Let the data sequence $T_{1:n} = \{Z_1, \ldots, Z_n\}$ be drawn from a stochastic process satisfying $\beta$-mixing, with mixing coefficients that satisfy $\sum_{k \geq 1} \beta(k) = B < \infty$. Let the loss function $\text{loss}(\cdot, \cdot)$ be bounded, and let $\sigma^2$ be an upper bound on its variance.*

*Then, for any $\delta \in (0, 1)$, with probability at least $1 - \delta$, the following inequality holds:*

$$G_n(A(T_{1:n})) \leq 2\varepsilon_{stab} + \sqrt{\frac{2\sigma^2(1 + 4B)\ln(2/\delta)}{n}} \tag{47}$$

*(Note: The constant $(1+4B)$ comes from tighter concentration inequalities for $\beta$-mixing sequences, such as variants of McDiarmid's or Bernstein's inequalities. The specific constant depends on the underlying concentration inequality being invoked.)*

*Proof.* Let $h = A(T_{1:n})$ denote the hypothesis (model) trained on the training set $T_{1:n}$. The generalization error is defined as the difference between the true risk and the empirical risk:

$$\begin{aligned} G_n(h) &= R(h) - R_{\text{emp}}(h) \\ &= \mathbb{E}_{Z \sim \mathcal{D}}[\text{loss}(h, Z)] - \frac{1}{n}\sum_{i=1}^{n}\text{loss}(h, Z_i). \end{aligned} \tag{48}$$

Our goal is to provide a high-probability upper bound for $G_n(h)$. We decompose the error into two parts: a **Bias Term** and a **Concentration Term**.

$$G_n(h) = \underbrace{(R(h) - \mathbb{E}[R_{emp}(h)])}_{\text{Bias Term}} + \underbrace{(\mathbb{E}[R_{emp}(h)] - R_{emp}(h))}_{\text{Concentration Term}} \tag{49}$$

By the triangle inequality, we can bound the two terms separately:

$$G_n(h) \leq |R(h) - \mathbb{E}[R_{emp}(h)]| + |\mathbb{E}[R_{emp}(h)] - R_{emp}(h)| \tag{50}$$

**Bounding the Bias Term** We first bound the term $|R(h) - \mathbb{E}[R_{emp}(h)]|$. The core of this step is to leverage the uniform stability of the algorithm. Through a classic symmetrization argument, which involves introducing a "ghost sample" drawn independently from the same distribution, it can be shown that uniform stability implies a bound on the gap between the true risk and the expected empirical risk:

$$|\mathbb{E}[R(h)] - \mathbb{E}[R_{emp}(h)]| \leq 2\varepsilon_{\text{stab}} \tag{51}$$

This bound is deterministic; it does not depend on a particular sample but only on the algorithm's stability property. It quantifies the systematic bias introduced because the algorithm uses the same data for both training and evaluation.

**Bounding the Concentration Term** Next, we bound the second term, $|R_{emp}(h) - \mathbb{E}[R_{emp}(h)]|$, which represents the deviation of the random variable $R_{emp}(h)$ from its expected value.

$$|R_{\text{emp}}(h) - \mathbb{E}[R_{\text{emp}}(h)]| = \left| \frac{1}{n}\sum_{i=1}^{n}\text{loss}(h, Z_i) - \mathbb{E}\left[\frac{1}{n}\sum_{i=1}^{n}\text{loss}(h, Z_i)\right] \right|. \tag{52}$$

Here, the randomness comes from the training data $T_{1:n}$. Since the sequence $\{Z_i\}$ is $\beta$-mixing, the sequence of random variables $\{loss(A(T_{1:n}), Z_i)\}$ is also a dependent sequence.

We can apply a concentration inequality designed for $\beta$-mixing sequences (e.g., a variant of Bernstein's or Hoeffding's inequality). For any $\gamma > 0$, such an inequality takes the form:

$$\Pr\left[|R_{emp}(h) - \mathbb{E}[R_{emp}(h)]| \geq \gamma\right] \leq 2\exp\left(-\frac{n\gamma^2}{C(\sigma^2, B)}\right) \tag{53}$$

where $C(\sigma^2, B)$ is a constant that depends on the variance upper bound $\sigma^2$ and the sum of mixing coefficients $B$. A common form is $C(\sigma^2, B) = 2\sigma^2(1 + 4B)$. Thus, we have:

$$\Pr\left[|R_{emp}(h) - \mathbb{E}[R_{emp}(h)]| \geq \gamma\right] \leq 2\exp\left(-\frac{n\gamma^2}{2\sigma^2(1 + 4B)}\right) \tag{54}$$

**Combining the Bounds** Now, we combine the results. We want the total error to be bounded with high probability, at least $1 - \delta$. From the concentration inequality in Step 2, we set the probability upper bound to $\delta$:

$$\delta = 2\exp\left(-\frac{n\gamma^2}{2\sigma^2(1 + 4B)}\right) \tag{55}$$

Solving for $\gamma$, we get the bound on the concentration term:

$$\gamma = \sqrt{\frac{2\sigma^2(1+4B)\ln(2/\delta)}{n}} \tag{56}$$

This means that, with probability at least $1-\delta$, we have:

$$|R_{emp}(h) - \mathbb{E}[R_{emp}(h)]| \leq \sqrt{\frac{2\sigma^2(1+4B)\ln(2/\delta)}{n}} \tag{57}$$

Combining this high-probability bound with the deterministic bound from Step 1, we obtain the final result:

$$G_n(h) \leq |R(h) - \mathbb{E}[R_{\text{emp}}(h)]| + |R_{\text{emp}}(h) - \mathbb{E}[R_{\text{emp}}(h)]|$$
$$\leq 2\varepsilon_{\text{stab}} + \sqrt{\frac{2\sigma^2(1+4B)\log(2/\delta)}{n}}. \tag{58}$$

This inequality holds with probability at least $1-\delta$. $\qquad\square$

## A.4 Non-Decreasing Task-Relevant Mutual Information

**Theorem A.7** (Patch Representation Improves the IB Objective under a Ratio Condition). *Assume the Markov chain $Y \leftrightarrow X \to Z_{point} \to Z_{patch}$, where $Z_{patch} = Q(Z_{point})$ is obtained from $Z_{point}$ by a (possibly randomized) quantization/patching operator $Q$ that does not look into the future of $X$. Define*

$$\Delta_X := I(X; Z_{point}) - I(X; Z_{patch}) \geq 0, \qquad \Delta_Y := I(Y; Z_{point}) - I(Y; Z_{patch}) \geq 0. \tag{59}$$

*Then for any $\beta > 0$,*

$$\mathcal{L}_{IB}(Z_{point}) - \mathcal{L}_{IB}(Z_{patch}) = \Delta_X - \beta\,\Delta_Y. \tag{60}$$

*In particular, using the definitions from equation 59,*

$$\mathcal{L}_{IB}(Z_{patch}) < \mathcal{L}_{IB}(Z_{point}) \quad \iff \quad \Delta_X > \beta\,\Delta_Y. \tag{61}$$

*Proof.* By definition $\mathcal{L}_{IB}(Z) = I(X; Z) - \beta\,I(Y; Z)$. Subtracting the two values at $Z_{\text{point}}$ and $Z_{\text{patch}}$ using the definitions from equation 59 gives equation 60 identically. The nonnegativity of $\Delta_X, \Delta_Y$ follows from the data processing inequality along the Markov chain $X \to Z_{\text{point}} \to Z_{\text{patch}}$ and $Y \to X \to Z_{\text{point}} \to Z_{\text{patch}}$. $\qquad\square \qquad\square$

**Corollary A.4** (A practical sufficient condition via denoising margin and MI preservation). *Suppose there exist constants $m > 0$ and $\eta \geq 0$ such that*

$$\Delta_X \geq m \qquad and \qquad \Delta_Y \leq \eta. \tag{62}$$

*Then for any $\beta \in \big(0,\ m/\eta\big)$, the condition in equation 61 is satisfied and one has $\mathcal{L}_{IB}(Z_{patch}) < \mathcal{L}_{IB}(Z_{point})$.*

**Remark A.12** (How to instantiate $m$ and $\eta$ in practice). *(a) Task-information preservation: If the quantizer is $\varepsilon$-faithful in input space (e.g., $\|Z_{point} - Z_\varepsilon\|_2 \leq \varepsilon$) and the classifier logits $f$ are $L_f$-Lipschitz, then by Theorem A.8 (a Lipschitz–Fannes bound), we can bound $\eta$ in equation 62:*

$$\Delta_Y = I(Y; Z_{point}) - I(Y; Z_{patch}) \leq C\,\varepsilon \quad (for\ small\ \varepsilon), \tag{63}$$

*with an explicit constant $C$ depending on $L_f$ and $|Y|$ (plus a $h_2(\cdot)$ term in the precise bound).*

*(b) Denoising margin on $X$: If patching/quantization removes at least $m$ nats of mutual information about $X$ (e.g., by enforcing a target distortion $D$ and using a rate–distortion upper bound $I(X; Z_{patch}) \leq R_X(D)$ so that $m \leq I(X; Z_{point}) - R_X(D)$), then $\Delta_X \geq m$.*

**Theorem A.8** ($\varepsilon$-MI Preservation via Lipschitz Continuity). *Let $Z_{pt}$ be the continuous (pointwise) representation and $Z_\varepsilon = Q_\varepsilon(Z_{pt})$ be its quantized version, satisfying $\|Z_{pt} - Z_\varepsilon\|_2 \leq \varepsilon$. Let the model be a probabilistic classifier where the conditional probability $P(Y|Z)$ is generated from a logit function $f(Z)$ followed by a softmax. Assume the logit function $f : \mathbb{R}^P \to \mathbb{R}^{|Y|}$ is $L_f$-Lipschitz continuous.*

*Then, the loss in mutual information is bounded:*

$$I(Y; Z_\varepsilon) \geq I(Y; Z_{pt}) - C \cdot \varepsilon \tag{64}$$

*where $C$ is a constant dependent on the model's Lipschitz constant and the number of task classes, for instance, $C = L_f \log(|Y| - 1)$ under certain tight bounding assumptions.*

Underlying Assumptions:

- **Bounded Quantization Error**: There exists a fixed $\varepsilon > 0$ such that for any $Z_{\text{pt}}$, we have $\|Z_{\text{pt}} - Q_\varepsilon(Z_{\text{pt}})\|_2 \leq \varepsilon$.
- **Probabilistic Model**: The model's conditional probability distribution $P(Y|Z)$ is generated by a softmax applied to a logit function, i.e., $P(Y|Z) = \text{softmax}(f(Z))$.
- **Model Smoothness**: The logit function $f$ is $L_f$-Lipschitz continuous with respect to the $L_2$ norm. This is a common assumption for robust models.

*Proof.* The proof proceeds by bounding the change in conditional entropy, which arises from the quantization error, through a chain of Lipschitz continuity arguments.

**Mutual Information Difference Decomposition.** We begin with the standard definition of mutual information, $I(Y;Z) = H(Y) - H(Y|Z)$. The difference can be expressed precisely as:

$$I(Y;Z_{\text{pt}}) - I(Y;Z_\varepsilon) = H(Y|Z_\varepsilon) - H(Y|Z_{\text{pt}}) = \mathbb{E}[H(Y|Z_\varepsilon)] - \mathbb{E}[H(Y|Z_{\text{pt}})] \tag{65}$$

Our goal is to find an upper bound for the right-hand side, which requires bounding the term $|H(Y|Z_\varepsilon) - H(Y|Z_{\text{pt}})|$.

**Bounding the Change in Conditional Entropy.** We establish a "continuity propagation chain" from the input representation $Z$ to the conditional entropy $H(Y|Z)$.

(a) *From Input to Logits:* By the Lipschitz assumption on the logit function $f$, the quantization error $\varepsilon$ bounds the change in the logits:

$$\|f(Z_\varepsilon) - f(Z_{\text{pt}})\|_2 \leq L_f \|Z_\varepsilon - Z_{\text{pt}}\|_2 \leq L_f \varepsilon. \tag{66}$$

(b) *From Logits to Probabilities (TV Distance):* The softmax function is also Lipschitz. It can be shown that the Total Variation (TV) distance between two output probability distributions is bounded by the difference in their input logits using equation 66.

$$\text{TV}(P_{Y|Z_\varepsilon}, P_{Y|Z_{\text{pt}}}) \leq \frac{1}{2}\|f(Z_\varepsilon) - f(Z_{\text{pt}})\|_1 \leq \frac{\sqrt{|Y|}}{2}\|f(Z_\varepsilon) - f(Z_{\text{pt}})\|_2 \leq \frac{\sqrt{|Y|}}{2}L_f \varepsilon. \tag{67}$$

This shows that a small perturbation in $Z$ leads to a proportionally small change in the conditional probability distribution.

(c) *From Probabilities to Entropy:* The entropy function $H(P) = -\sum p_i \log p_i$ is Lipschitz continuous over the probability simplex. Its Lipschitz constant, $L_H$, with respect to the TV distance (or L1 norm), can be bounded, e.g., by $L_H \leq \log(|Y| - 1)$. Thus, the change in entropy is bounded by the change in the probability distribution:

$$|H(Y|Z_\varepsilon) - H(Y|Z_{\text{pt}})| = |H(P_{Y|Z_\varepsilon}) - H(P_{Y|Z_{\text{pt}}})| \leq L_H \cdot \text{TV}(P_{Y|Z_\varepsilon}, P_{Y|Z_{\text{pt}}}). \tag{68}$$

**Combining the Bounds.** By chaining the inequalities from equation 67 and equation 68, we get a direct bound on the change in entropy for any given point:

$$|H(Y|Z_\varepsilon) - H(Y|Z_{\text{pt}})| \leq \log(|Y| - 1) \cdot \frac{\sqrt{|Y|}}{2} L_f \varepsilon. \tag{69}$$

Let's define the constant $C = L_f \log(|Y| - 1)\frac{\sqrt{|Y|}}{2}$ (or a tighter version thereof). We have $|H(Y|Z_\varepsilon) - H(Y|Z_{\text{pt}})| \leq C \cdot \varepsilon$.

Returning to the mutual information difference in equation 65, we take the expectation over all possible values. By linearity of expectation and Jensen's inequality:

$$\begin{aligned}
|I(Y;Z_{\text{pt}}) - I(Y;Z_\varepsilon)| &= |\mathbb{E}\left[H(Y|Z_\varepsilon) - H(Y|Z_{\text{pt}})\right]| \\
&\leq \mathbb{E}\left[|H(Y|Z_\varepsilon) - H(Y|Z_{\text{pt}})|\right] \\
&\leq \mathbb{E}[C \cdot \varepsilon] = C \cdot \varepsilon.
\end{aligned} \tag{70}$$

This yields the final result from equation 70, $I(Y;Z_{\text{pt}}) - I(Y;Z_\varepsilon) \leq C \cdot \varepsilon$, which can be rewritten as:

$$I(Y;Z_\varepsilon) \geq I(Y;Z_{\text{pt}}) - C \cdot \varepsilon. \tag{71}$$

$\square$

## B DATASETS OVERVIEW

This study utilizes a comprehensive collection of 37 time series datasets spanning multiple domains, totaling approximately 187 million data points across diverse temporal scales. The datasets encompass various temporal resolutions from 4-second ultra-high-frequency data to monthly observations, providing comprehensive coverage for time series analysis and forecasting tasks across energy, finance, transportation, environmental, and healthcare domains.

Table 3: Overview of Time Series Datasets

| Dataset | Length | Features | Total Points | Size (MB) | Frequency | Description |
|---|---|---|---|---|---|---|
| ETTh1 | 17,420 | 7 | 121,940 | 0.93 | 1 hour | Power transformer temperature (2016.7-2018.7) |
| ETTh2 | 17,420 | 7 | 121,940 | 0.93 | 1 hour | Power transformer temperature (2016.7-2018.7) |
| ETTm1 | 69,680 | 7 | 487,760 | 3.72 | 15 min | Power transformer temperature (2016.7-2018.7) |
| ETTm2 | 69,680 | 7 | 487,760 | 3.72 | 15 min | Power transformer temperature (2016.7-2018.7) |
| NASDAQ | 14,224 | 1 | 14,224 | 0.11 | 1 day | NASDAQ composite index historical data |
| SP500 | 2,598 | 1 | 2,598 | 0.02 | 1 day | Standard & Poor's 500 index data |
| WTH | 35,064 | 12 | 420,768 | 3.21 | 1 hour | Weather data (12 meteorological parameters) |
| Air Quality | 9,357 | 13 | 121,641 | 0.93 | 1 hour | Air quality monitoring (13 environmental indicators) |
| COVID Deaths | 212 | 233 | 49,396 | 0.38 | 1 day | COVID-19 death cases time series data |
| Electricity | 26,304 | 321 | 8,443,584 | 64.42 | 1 hour | Electricity load (321 consumption points) |
| Electricity Demand | 230,736 | 5 | 1,153,680 | 8.8 | 1 min | High-frequency electricity demand (4+ years) |
| Electricity Weekly | 156 | 321 | 50,076 | 0.38 | 1 week | Electricity load (weekly aggregated) |
| Energy | 19,735 | 26 | 513,110 | 3.91 | 1 day | Energy consumption (26 indicators) |
| Exchange Rate | 7,588 | 8 | 60,704 | 0.46 | 1 day | Foreign exchange rates (8 currency pairs) |
| FRED-MD | 728 | 107 | 77,896 | 0.59 | 1 month | Federal Reserve economic data (107 indicators) |
| Hospital | 84 | 767 | 64,428 | 0.49 | 1 week | Hospital-related time series (767 indicators) |
| Web Traffic Daily | 803 | 145,035 | 116,463,105 | 888.54 | 1 day | Wikipedia page views (145,035 pages) |
| Web Traffic Weekly | 114 | 145,035 | 16,533,990 | 126.14 | 1 week | Wikipedia page views (weekly aggregated) |
| Metro | 48,204 | 1 | 48,204 | 0.37 | 1 hour | Urban metro passenger traffic |
| National Illness | 966 | 7 | 6,762 | 0.05 | 1 week | Disease surveillance (7 illness types) |
| NN5 Daily | 791 | 111 | 87,801 | 0.67 | 1 day | NN5 competition dataset (111 series) |
| NN5 Weekly | 113 | 111 | 12,543 | 0.1 | 1 week | NN5 competition dataset (weekly) |
| Oikolab Weather | 100,057 | 8 | 800,456 | 6.11 | 1 hour | Oikolab weather data (8 parameters) |
| River Flow | 23,741 | 1 | 23,741 | 0.18 | 1 day | River discharge monitoring |
| Saugeen Day | 23,741 | 1 | 23,741 | 0.18 | 1 day | Saugeen river basin data |
| Solar 10min | 52,560 | 137 | 7,200,720 | 54.94 | 10 min | Solar power (137 stations, Alabama 2006) |
| Solar 1min | 493,149 | 1 | 493,149 | 3.76 | 1 min | High-frequency solar power generation |
| Solar 4sec | 7,397,222 | 1 | 7,397,222 | 56.44 | 4 sec | Ultra-high-frequency solar power data |
| Solar Weekly | 52 | 137 | 7,124 | 0.05 | 1 week | Solar power generation (weekly) |
| Sunspot | 73,924 | 1 | 73,924 | 0.56 | 1 month | Solar activity data (1749-present, 270+ years) |
| TCPC | 52,416 | 8 | 419,328 | 3.2 | 1 hour | Temperature-correlated power consumption |
| Traffic | 17,544 | 862 | 15,122,928 | 115.38 | 1 hour | Highway traffic (862 monitoring points) |
| Traffic Weekly | 104 | 862 | 89,648 | 0.68 | 1 week | Traffic flow (weekly aggregated) |
| US Births | 7,305 | 1 | 7,305 | 0.06 | 1 day | US birth rate data |
| Weather | 52,696 | 21 | 1,106,616 | 8.44 | 1 hour | Comprehensive weather (21 parameters) |
| Wind 1min | 493,144 | 1 | 493,144 | 3.76 | 1 min | High-frequency wind power generation |
| Wind 4sec | 7,397,147 | 1 | 7,397,147 | 56.44 | 4 sec | Ultra-high-frequency wind power |

### B.1 DATASET CHARACTERISTICS AND CLASSIFICATION

Our comprehensive dataset collection exhibits diverse temporal scales and application domains, providing robust coverage for time series foundation model analysis. The datasets are systematically categorized across multiple dimensions:

#### B.1.1 TEMPORAL SCALE CLASSIFICATION

**Ultra-High Frequency (Second-Level):** 3 datasets featuring 4-second sampling intervals, primarily capturing renewable energy fluctuations and ultra-short-term dynamics essential for grid stability analysis.

**High Frequency (Minute-Level):** 4 datasets with 1-minute to 15-minute resolution, including power generation monitoring and device sensor measurements, enabling fine-grained pattern analysis.

**Medium Frequency (Hourly):** 12 datasets providing hourly observations across energy, environmental, and transportation domains, representing the most common temporal resolution for operational forecasting.

**Low Frequency (Daily and Above):** 21 datasets spanning daily, weekly, and monthly intervals, covering long-term trends in financial markets, economic indicators, and natural phenomena.

### B.1.2 DATA SCALE CLASSIFICATION

**Ultra-Large Scale (>100M data points):** 1 dataset - Wikipedia web traffic with 116M observations across 145,035 pages, representing massive-scale digital behavior patterns.

**Large Scale (10M-100M data points):** 5 datasets including electricity grid monitoring and ultra-high-frequency renewable energy measurements, totaling over 50M observations.

**Medium Scale (100K-10M data points):** 15 datasets covering diverse applications from traffic monitoring to weather systems, providing substantial data for pattern discovery.

**Small Scale (<100K data points):** 19 datasets including financial indices, disease surveillance, and specialized monitoring systems, offering focused domain expertise.

### B.1.3 DOMAIN DISTRIBUTION

**Energy and Power (12 datasets):** Comprehensive coverage of electricity consumption, renewable generation (solar, wind), and power infrastructure monitoring, representing critical infrastructure systems.

**Finance and Economics (3 datasets):** Stock indices (NASDAQ, S&P 500), currency exchange rates, and macroeconomic indicators (FRED-MD), capturing market dynamics and economic trends.

**Environmental Monitoring (4 datasets):** Weather systems, air quality measurements, and atmospheric conditions, providing environmental pattern analysis capabilities.

**Transportation Systems (4 datasets):** Highway traffic flows and urban metro passenger data, enabling mobility pattern understanding and infrastructure optimization.

**Healthcare and Demographics (3 datasets):** Disease surveillance, hospital operations, and birth rate statistics, supporting public health analysis and demographic forecasting.

**Digital Systems (2 datasets):** Web traffic patterns capturing online behavior dynamics across massive digital platforms.

**Natural Phenomena (4 datasets):** River flows and solar activity measurements, representing long-term natural cycles and environmental processes.

**Specialized Applications (8 datasets):** Competition datasets (NN5), research-specific measurements, and domain-specific monitoring systems.

## B.2 DATASET COLLECTION STATISTICS

The complete dataset ensemble provides unprecedented scale and diversity:

- **Total data points:** Approximately 200 million observations
- **Temporal coverage:** 4-second to monthly sampling intervals
- **Dimensionality spectrum:** 1 to 145,035 features per dataset
- **Domain coverage:** 8 distinct application sectors
- **Storage requirements:** Over 1.3 GB of time series data
- **Temporal span:** From ultra-short-term (seconds) to multi-decade (270+ years)

This comprehensive collection enables systematic evaluation of time series foundation models across diverse temporal patterns, scales, and application domains, supporting robust cross-domain generalization analysis and pattern recognition research.

## C THE GRAMMAR OF TIME SERIES

Having established that time series can be tokenized into a robust motif "vocabulary" exhibiting language-like frequency regularities, we now ask a deeper question: do these motifs concatenate arbitrarily, or do they obey a discernible **grammar**? A language is defined not only by its words

but by the rules that govern their composition. Using transition and $n$-gram analyses, we provide quantitative evidence that patch-token sequences are governed by a clear, non-trivial grammar.

## C.1 N-GRAM ANALYSIS

To investigate grammatical structure in time series, we employ **n-gram analysis**, a fundamental technique from computational linguistics that examines sequences of $n$ consecutive elements (tokens) to reveal patterns of co-occurrence and sequential dependencies. In natural language, n-grams capture local syntax: 2-grams (bigrams) reveal word pairs like "neural network," while 3-grams (trigrams) capture phrases like "machine learning algorithms." Applied to our time series context, n-grams of patch tokens reveal temporal motif sequences that may indicate underlying dynamical rules governing system evolution.

Our analysis focuses on several key dimensions: **(1) Transition probabilities** between consecutive tokens (2-grams), revealing preferred motif sequences and state persistence; **(2) Coverage patterns** showing how sequence complexity scales with n-gram length; and **(3) Hierarchical structure** indicating whether longer patterns exhibit compositional properties analogous to linguistic phrases.

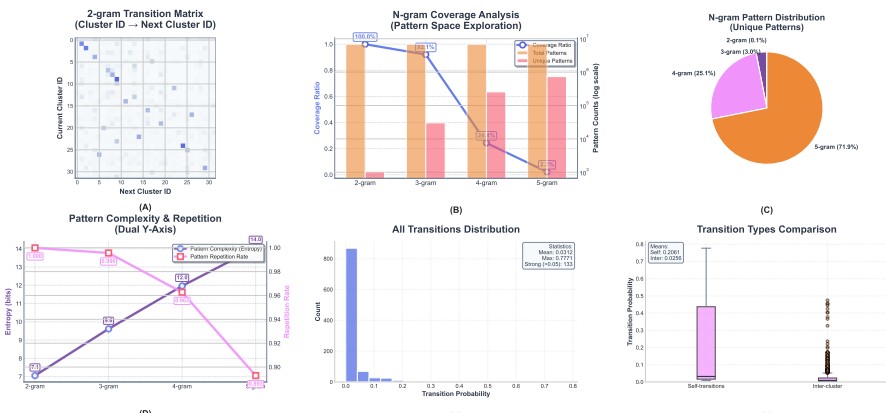

Figure 6: **Comprehensive grammatical analysis of patch–token time–series sequences (32 clusters).** **(A)** *2-gram transition matrix*: Row–normalized $32 \times 32$ probabilities showing dominant diagonal (*state inertia*) and high–probability off–diagonal pairs (*preferred transitions*). **(B)** *N-gram coverage & counts*: For $n \in \{2, 3, 4, 5\}$, coverage ratio drops, with log-scaled bars showing total vs. unique pattern counts. **(C)** *Composition of unique patterns*: 5-grams dominate, followed by 4-grams, indicating rich "phrases." **(D)** *Complexity vs. repetition*: Pattern entropy increases while repetition rate decreases, supporting *chunking* hierarchy. **(E)** *Transition probability distribution*: Most edges are near zero with a long tail. **(F)** *Self vs. inter-cluster transitions*: Box plots show self-transitions are substantially larger than inter-cluster ones, quantifying inertia and the sparsity of cross-state moves.

**The Principle of State Inertia.** Our primary discovery, clearly visible in the State Transition Probability Matrix (Panel A), is the overwhelming dominance of self-transitions. The bright diagonal line indicates that once a particular temporal motif is established, it has a very high probability of persisting in the subsequent time step. This principle of state inertia represents the simplest and most powerful rule of temporal grammar: *dynamics are persistent. This phenomenon mirrors structures in natural language that maintain focus—just as a paragraph typically revolves around a core topic, keeping its conceptual 'state' persistent, or as a series of adjectives describe a single noun (e.g., "a long, dark, quiet road"), the temporal 'subject' (i.e., the current dynamic pattern) tends to endure across time steps.*

**A Highly Structured and Sparse Language.** While motifs tend to repeat, their transitions to different motifs are far from random. The space of "grammatically valid" motif combinations is extremely sparse, as illustrated by the exponential decay in N-gram Coverage Analysis (Panel B). While 100% of 1-grams (single motifs) are observed, the coverage drops precipitously for higher-order n-grams, indicating that only a small fraction of all possible motif sequences are "grammat-

ically correct" or physically plausible. *This provides the most direct parallel to natural language grammar—just as English syntax dictates that 'The cat sat' is valid while 'Sat the cat' is not, the grammar of time permits only a highly structured subset of all possible motif combinations.* This sparsity constitutes strong evidence for powerful, underlying compositional rules.

**Rich Temporal Phraseology.** The analysis of higher-order n-grams reveals a sophisticated temporal vocabulary structure. The predominance of 5-grams and 4-grams in unique pattern composition (Panel C) indicates that meaningful temporal structures naturally span multiple time steps, forming coherent "temporal phrases." These longer sequences, while individually less frequent, collectively represent a rich vocabulary of complex temporal expressions. *This hierarchical organization mirrors natural language, where phrases convey more semantic content than individual words, and where the expressive power emerges from compositional complexity rather than mere token frequency.*

**Macroscopic Diversity from Microscopic Chunking.** At first glance, the dominance of self-loops might suggest monotonous sequences. However, the relationship between pattern complexity and repetition rates (Panel D) reveals a more nuanced reality: sequences exhibit high macroscopic diversity through a "chunking" mechanism. This apparent paradox is resolved by understanding that complex sequences are constructed by composing persistent chunks of motifs—a typical sequence is not uniformly simple or complex, but rather a concatenation of internally-stable segments that generates high overall diversity and uniqueness. *This perfectly mirrors the hierarchical structure of natural language, where complex sentences are structured compositions of well-defined phrases and clauses, allowing immense expressive diversity while adhering to simpler local rules.*

**Long-tail Grammatical Distribution.** The distribution of transition probabilities (Panel E) follows a characteristic long-tail pattern fundamental to natural language systems. Most motif transitions exhibit very low probabilities, while only a select few achieve high-probability status. *This structure demonstrates that temporal sequences, like natural languages, are governed by usage-based statistical regularities where a small set of "preferred expressions" dominates actual occurrence, while the vast majority of theoretically possible combinations remain rare or nonexistent.* This distribution pattern provides compelling evidence that time series data exhibits genuine linguistic properties in its compositional structure.

### C.2 TEMPORAL VOCABULARY ANALYSIS

To qualitatively understand the vocabulary discovered by our data-driven approach, Figure 7 visualizes the complete set of 32 cluster centroids, or "temporal motifs", learned from the dataset. While arranged by cluster ID, each motif's frequency of occurrence reveals the structural composition of the "language of time". Note that we use full dataset with $P = 32$, rather than sampling.

**A Hierarchy from Simple States to Complex Events.** A striking feature revealed in Figure 7 is the emergent hierarchy of pattern complexity based on frequency distribution. The most dominant motif is Cluster 1 (16,710,261 samples, 18.9%), representing the fundamental near-zero stable state that forms the baseline of most time series. This is followed by Cluster 1 (14,461,985 samples, 16.3%), showing a almost-zero stable state, and Cluster 13 (9,852,111 samples, 11.1%), representing a pattern with slight fluctuations. Together, these top three clusters account for over 46% of all temporal patterns, establishing the foundational vocabulary of time series dynamics.

The frequency hierarchy reveals distinct pattern categories with clear functional roles:

- **Dominant Base States (>5% each):** Clusters 1, 9, 13, 6, 20 fundamental patterns and simple trends, representing stable states, slight fluctuations, and obvious periodic fluctuations, respectively. collectively comprising 57.4% of all motifs. These act as the "grammatical foundation" of temporal language.

- **Common Dynamic Patterns (1-5%):** Clusters like 4, 25, 29 show moderate-frequency transitional and stable states, providing structural diversity.

- **Rare Complex Events (<1%):** The least frequent motifs (Clusters 23, 30, 15 at the bottom of the hierarchy) represent specialized, high-amplitude patterns and complex transitions that capture exceptional temporal behaviors.

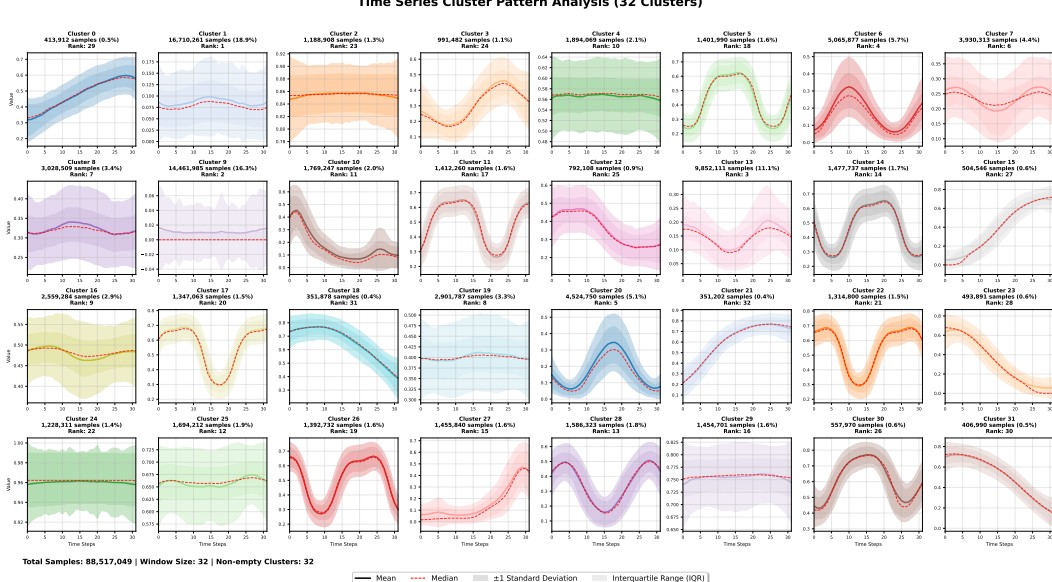

Figure 7: **The Learned Vocabulary of Temporal Motifs: Visualizing the 32 Cluster Centers.**
This figure displays the 32 cluster centers, or 'temporal motifs,' learned by the K-Means algorithm
(K=32) from time series patches of length 32. Each plot represents a single prototypical pattern.
The plots are arranged by cluster ID (from Cluster 0 to Cluster 31), with each cluster's frequency
of occurrence (cluster size, denoted by $n$) indicated in the subplot titles. Additionally, each cluster
visualization includes statistical summaries showing the mean, median, one standard deviation, and
quartile intervals.

This frequency-complexity relationship mirrors natural language structure, where simple grammat-
ical elements dominate usage while rare, semantically rich patterns convey specific temporal events
and anomalous behaviors.

**Qualitative Validation of the Linguistic Analogy.** The structure of this learned vocabulary pro-
vides strong qualitative validation for our central hypothesis. The inverse relationship between pat-
tern complexity and frequency—whereby simple, foundational patterns are ubiquitous and complex,
event-specific patterns are rare—aligns perfectly with the quantitative findings of Zipf's Law pre-
sented in our earlier analysis. The ability to automatically discover such a rich, interpretable, and
comprehensive lexicon from raw data demonstrates that complex time series dynamics are indeed
compositional. This confirms that a finite set of reusable motifs forms the basis of observed signals,
providing a solid foundation for treating time series analysis as a language modeling task.

## C.3 TRANSITION & CO-OCCURRENCE PATTERNS

We conducted comprehensive 2-gram and 3-gram analyses on the sampled patches, systematically
investigating n-gram transition patterns between temporal motifs. This analysis revealed several
intriguing phenomena that provide deeper insights into the grammatical structure of time series
data.

**State Inertia and Transition Dynamics**. The n-gram analysis reveals distinct behavioral patterns
between high-frequency and low-frequency motifs. High-frequency motifs exhibit pronounced state
inertia (stickiness), demonstrating strong self-loop tendencies that preserve temporal stability. Con-
versely, low-frequency motifs display substantially reduced self-transition frequencies, reflecting
their specialized roles in capturing transient dynamics and anomalous temporal behaviors. This di-
chotomy arises from the fundamental stationarity characteristics of real-world time series, where
dominant stable motifs maintain sequence coherence through self-reinforcing transitions.

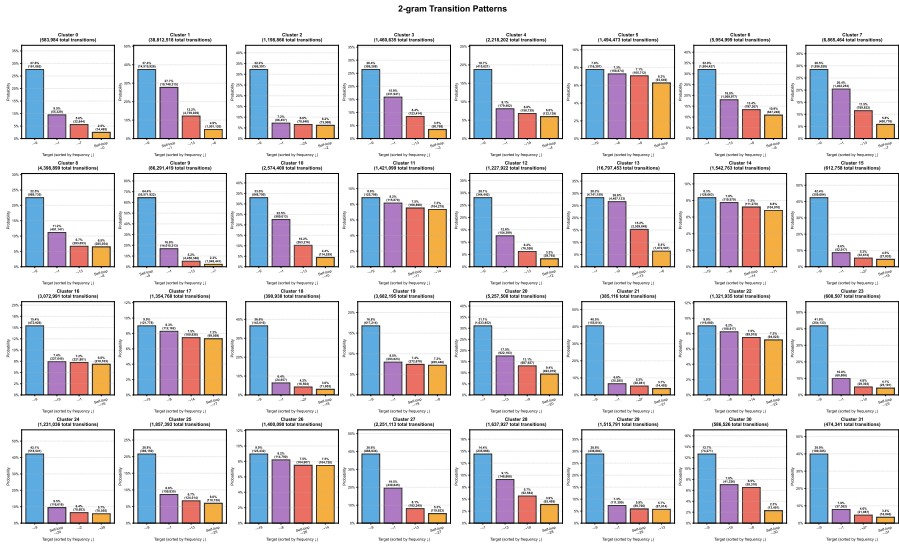

Figure 8: **2-gram Transition Pattern Analysis.** This figure displays the highest-frequency outer-cluster transitions from each cluster, along with self-loop transition frequencies. The analysis reveals the predominant transition patterns between temporal motifs, highlighting both inter-cluster relationships and state persistence behavior. **Bars with bold borders represent self-loop transitions.**

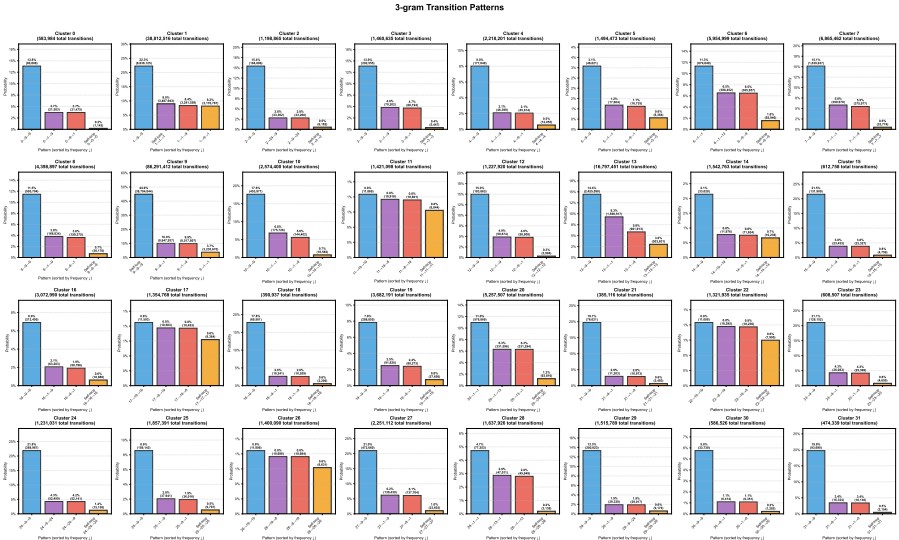

Figure 9: **3-gram Transition Pattern Analysis.** This figure illustrates higher-order 3-gram transition patterns, showing the most frequent outer-cluster transitions and self-loop frequencies from each cluster. The analysis captures complex temporal phrase structures and hierarchical organization in motif sequences. **Bars with bold borders represent self-loop transitions.**

**Directional Flow Toward Equilibrium States**  . Our analysis reveals a systematic directional bias in motif transitions: low-frequency patterns exhibit preferential transition pathways toward high-frequency base states. This asymmetric flow pattern—from specialized dynamic patterns and rare complex events toward dominant equilibrium states—reflects a fundamental thermodynamic principle governing temporal systems. Without sustained external perturbation, dynamic systems naturally evolve toward configurations of minimum energy and maximum entropy, corresponding to stable or equilibrium states. This transition bias provides empirical evidence for the physical interpretability of motif hierarchies, where rare transient patterns serve as departure mechanisms from, and eventual return pathways to, the system's preferred steady states.

**Stationarity Dominance**  . A fundamental insight emerges from our comprehensive n-gram analysis: across both 2-gram and 3-gram patterns, time series motifs predominantly manifest in stationary configurations. This observation reveals a critical characteristic of temporal "grammar"—time series inherently favor stable, predictable patterns as their default linguistic mode. Non-stationary phenomena occur only under exceptional and sporadic circumstances, representing rare deviations from the underlying equilibrium structure. This grammatical bias toward stationarity suggests that temporal sequences, much like natural languages, possess an intrinsic preference for stable syntactic patterns, with instability serving as a marked linguistic feature reserved for specific contextual or anomalous conditions. Moreover, this stationarity dominance exhibits a hierarchical organization: high-frequency motifs establish the foundational grammatical layer through consistent self-reinforcement, while progressively lower-frequency patterns form ascending hierarchical levels that encode increasingly specialized and context-dependent temporal structures, creating a stratified linguistic architecture where stability decreases systematically with hierarchical depth.

The preceding analysis reveals that motif transitions and co-occurrence patterns in time series exhibit remarkable predictability and structural simplicity. This observation directly validates our theoretical framework: patch-based time series modeling fundamentally decomposes forecasting into two tractable sub-problems—*motif pattern classification* and *continuous offset regression*. The former leverages the highly predictable transition dynamics demonstrated in our n-gram analysis, where stable motifs exhibit strong self-loop tendencies and rare patterns follow systematic return pathways to equilibrium states. The latter exploits the inherent smoothness of numerical deviations within motif neighborhoods, where local variations follow predictable distributional patterns. This decomposition mechanism constitutes a fundamental advantage of patch-based approaches: by transforming complex temporal forecasting into discrete pattern recognition combined with continuous fine-tuning, patches naturally align with the underlying grammatical structure of time series, enabling superior representational capacity and cross-domain generalization capabilities.

## C.4 PATCH-LEVEL MOTIF VISUALIZATION

To elucidate the practical efficacy of our patch-based framework, we present a comprehensive visualization analysis using real-world time series data. Our methodology involved randomly selecting five time series sequences, each spanning 512 time points, and systematically partitioning them into patches using a fixed window length of 32. For each resulting patch, we mapped motif cluster assignments and visualized the statistical distribution characteristics of the corresponding cluster data.

For each identified motif cluster, we calculated three essential statistical measures across all constituent data points: the mean, median, and one-standard-deviation distribution ranges. This analytical framework serves two fundamental objectives: (1) it empirically validates the clustering efficacy of our motif identification approach, demonstrating how temporally similar patterns are systematically grouped into coherent categories; (2) it reveals the intrinsic statistical properties within each motif cluster, offering quantitative evidence for the structural coherence and stability of the identified temporal patterns.

The visualization results demonstrate our patch-based decomposition methodology in practice, wherein complex time series are hierarchically segmented into interpretable temporal units (patches), each distinguished by its dominant motif cluster and bounded by a statistical envelope defined by the cluster's central tendencies and dispersion characteristics. This analytical framework successfully integrates discrete pattern classification with continuous value regression, establishing the conceptual foundation for our patch-based modeling paradigm, as illustrated in Figure 10.

Our tokenization approach employs k-means clustering to generate vocabulary and utilizes KNN for predicting class cluster (motif set) membership. Overall, this method is simple and efficient, yet insufficient to demonstrate the optimal tokenization scheme. Similar to LLMs, different tokenization approaches often yield varying results.

As a comparative case, we incorporated a $K = 512$ setting, which expands the vocabulary to 512 motif sets with each pattern characterized by greater precision and specificity to particular motif types, as visualized in Figure 11. This comparison further illustrates that our vocabulary is not as deterministically fixed as in traditional NLP, but rather exhibits distinct characteristics under different configurations. This flexibility stems from our "token distributionalization" design, which introduces controlled randomness into the tokenization process.

Nevertheless, the underlying principles behind different schemes should remain consistent.

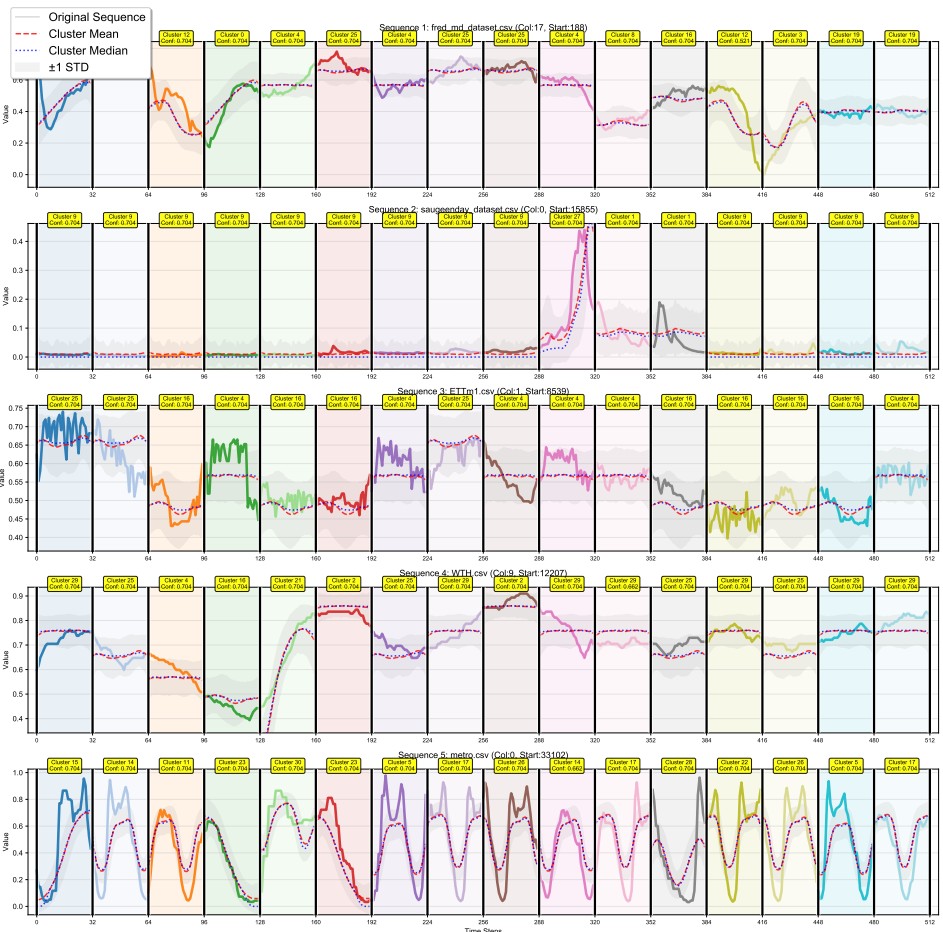

Figure 10: **Patch-Level Motif Cluster Analysis with Statistical Envelopes** ($K = 32$). This visualization demonstrates the practical application of our patch-based framework on five randomly selected time series sequences (each of length 512). Each time series is partitioned into patches using a window length of 32, with each patch colored according to its dominant motif cluster assignment. For each motif cluster, we display the statistical envelope defined by the cluster's mean (solid line), median (dashed line), and one-standard-deviation boundaries (shaded regions). This analysis reveals how similar temporal patterns are grouped into coherent clusters and demonstrates the inherent statistical properties within each motif class.

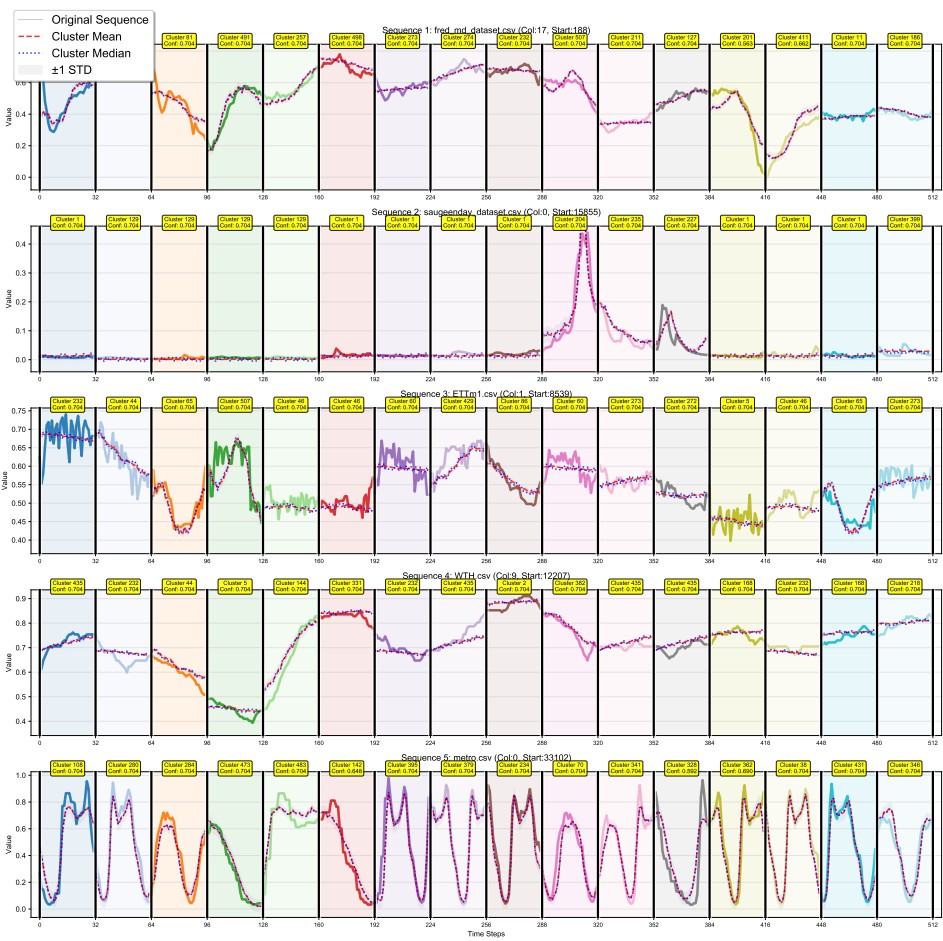

Figure 11: **Patch-Level Tokenization Visualization Comparison** ($K = 512$). This figure presents the $K = 512$ version of Figure 10, demonstrating tokenization results on specific time series with increased vocabulary size and motif specificity. When the vocabulary is larger ($K = 512$), an intuitive advantage is that each motif set provides more fine-grained characterization of corresponding patches, but this still needs to face the trade-off of over-fitting.

# D    METRICS FOR PATCH SIZE EVALUATION

This appendix provides comprehensive mathematical formulations and implementation details for the four core statistical metrics used to quantitatively evaluate patch size performance in time series analysis. These metrics address the fundamental question of why patch=1, despite achieving high clustering scores, performs poorly in temporal analysis tasks.

## D.1    TEMPORAL CONTEXT SCORE

### D.1.1    OBJECTIVE

Measures how well patches preserve temporal dependencies and sequential relationships inherent in time series data.

### D.1.2    CORE PRINCIPLE

- **patch length** $P = 1$: Single time points cannot capture temporal dependencies, resulting in zero score

- **patch length** $P > 1$: Temporal continuity assessed through autocorrelation analysis

### D.1.3    IMPLEMENTATION ALGORITHM

1. **Data Preprocessing:** Randomly sample min(1000, N) patches for computational efficiency

2. **Autocorrelation Computation:** For each patch with length > 1:

$$\text{normalized\_patch} = \frac{\text{patch} - \bar{x}}{\sigma} \tag{72}$$

$$\rho_{lag1} = \text{corr}(\text{patch}_{1:k-1}, \text{patch}_{2:k}) \tag{73}$$

3. **Score Aggregation:** $S_{TCP} = \frac{1}{N} \sum_{i=1}^{N} |\rho_{i,lag1}|$

### D.1.4    MATHEMATICAL FORMULATION

$$S_{TCP} = \begin{cases} 0 & \text{if } k = 1 \\ \frac{1}{N} \sum_{i=1}^{N} \left| \frac{\sum_{t=1}^{k-1}(x_{i,t}-\bar{x}_i)(x_{i,t+1}-\bar{x}_i)}{(k-1)\cdot\sigma_i^2} \right| & \text{if } k > 1 \end{cases} \tag{74}$$

### D.1.5    INTERPRETATION

- **High Score**: Strong correlations between consecutive time points, preserving temporal structure

- **Low Score**: Lack of temporal continuity, loss of sequential information

- **Theoretical Range**: [0, 1], with patch=1 guaranteed to be 0

## D.2    SEMANTIC CONSISTENCY

### D.2.1    OBJECTIVE

Quantifies the internal pattern consistency and smoothness within patches, measuring how well patches maintain coherent semantic structure.

### D.2.2    CORE PRINCIPLE

Utilizes Savitzky-Golay filtering to identify underlying patterns and computes the variance ratio between smoothed and original signals to assess pattern dominance over noise.

### D.2.3 IMPLEMENTATION ALGORITHM

1. **Special Case Handling:**

$$S_{SC} = 0.1 \quad \text{if } k = 1 \text{ (minimal semantic consistency)} \tag{75}$$

2. **Savitzky-Golay Smoothing:** For patches with length $P > 2$:

$$\text{window\_length} = \min(5, \lfloor k/2 \rfloor \times 2 + 1) \tag{76}$$
$$\hat{x}_i = \text{savgol\_filter}(x_i, \text{window\_length}, \text{poly\_order} = 2) \tag{77}$$

3. **Variance Ratio Computation:**

$$\text{consistency}_i = \frac{\text{Var}(\hat{x}_i)}{\text{Var}(x_i)} \tag{78}$$

4. **Fallback Method:** When smoothing fails, use first-order difference approach:

$$\text{consistency}_i = 1 - \frac{\text{Var}(\Delta x_i)}{\text{Var}(x_i)} \tag{79}$$

where $\Delta x_{i,t} = x_{i,t+1} - x_{i,t}$

5. **Final Aggregation:** $S_{SC} = \frac{1}{N} \sum_{i=1}^{N} \text{consistency}_i$

### D.2.4 MATHEMATICAL FORMULATION

$$S_{SC} = \begin{cases} 0.1 & \text{if } k = 1 \\ \frac{1}{N} \sum_{i=1}^{N} \frac{\text{Var}(\text{savgol\_filter}(x_i))}{\text{Var}(x_i)} & \text{if smoothing succeeds} \\ \frac{1}{N} \sum_{i=1}^{N} \left(1 - \frac{\text{Var}(\Delta x_i)}{\text{Var}(x_i)}\right) & \text{if smoothing fails} \end{cases} \tag{80}$$

### D.2.5 INTERPRETATION

- **High Score**: Patches exhibit smooth, consistent patterns with minimal irregular noise
- **Low Score**: Patches contain erratic, inconsistent variations
- **Ideal Scenario**: Underlying patterns dominate over random noise fluctuations

## D.3 NOISE SENSITIVITY

### D.3.1 OBJECTIVE

Assesses patch robustness against noise contamination, with lower sensitivity indicating better performance in noisy environments.

### D.3.2 CORE PRINCIPLE

Based on signal averaging theory where larger patches benefit from $\sqrt{N}$ noise reduction effect through statistical averaging of independent noise samples.

### D.3.3 IMPLEMENTATION ALGORITHM

1. **Base Noise Level Estimation:**

$$\sigma_{\text{base}} = 0.2 \times \text{std}(\text{sample\_data}) \tag{81}$$

2. **Theoretical Noise Reduction:**

$$\sigma_{\text{effective}} = \frac{\sigma_{\text{base}}}{\sqrt{k}} \tag{82}$$

*Mathematical Rationale:* Averaging N independent noise samples reduces noise variance by factor N, hence noise standard deviation by $\sqrt{N}$.

3. **Monte Carlo Empirical Validation:** For 50 trials:

$$\epsilon \sim \mathcal{N}(0, \sigma_{\text{effective}}^2) \tag{83}$$

$$x^{\text{noisy}} = x + \epsilon \tag{84}$$

4. **Quality Assessment:**
   - **For $P = 1$:** Signal-to-noise power ratio: $Q = \frac{P_{\text{signal}}}{P_{\text{signal}} + P_{\text{noise}}}$
   - **For $P > 1$:** Correlation preservation: $Q = \text{corr}(x_{\text{original}}, x_{\text{noisy}})$

5. **Theoretical Sensitivity:**

$$S_{\text{theoretical}} = 1.0 - \frac{\log_2(k+1)}{\log_2(129)} \tag{85}$$

6. **Empirical Sensitivity:** $S_{\text{empirical}} = 1.0 - \overline{Q}$

7. **Combined Score:** $S_{NS} = 0.3 \times S_{\text{empirical}} + 0.7 \times S_{\text{theoretical}}$

### D.3.4 INTERPRETATION

- **Low Sensitivity**: Patch robust against noise, maintains signal quality in contaminated environments
- **High Sensitivity**: Patch easily corrupted by noise, signal quality rapidly degrades
- **General Trend**: Larger patches exhibit lower sensitivity (greater robustness)

## D.4 PATTERN COMPLEXITY

### D.4.1 OBJECTIVE

Quantifies the richness and diversity of patterns captured within patches, measuring the ability to encode complex temporal features beyond simple monotonic trends.

### D.4.2 CORE PRINCIPLE

Combines multiple pattern features (trends, variations, second-order derivatives) to characterize pattern complexity comprehensively.

### D.4.3 IMPLEMENTATION ALGORITHM

1. **Special Case Handling:**

$$S_{PC} = 0.05 \quad \text{if } k = 1 \text{ (minimal pattern complexity)} \tag{86}$$

2. **Basic Statistical Features:** For each patch:

$$\mu_i = \frac{1}{k} \sum_{t=1}^{k} x_{i,t} \tag{87}$$

$$\sigma_i = \sqrt{\frac{1}{k} \sum_{t=1}^{k} (x_{i,t} - \mu_i)^2} \tag{88}$$

3. **Trend Analysis:** Linear trend slope extraction:

$$\beta_i = \frac{k \sum_{t=1}^{k} t \cdot x_{i,t} - \sum_{t=1}^{k} t \sum_{t=1}^{k} x_{i,t}}{k \sum_{t=1}^{k} t^2 - \left(\sum_{t=1}^{k} t\right)^2} \tag{89}$$

4. **Variation Pattern Analysis:** Second-order difference for curvature:

$$\Delta^2 x_{i,t} = x_{i,t+2} - 2x_{i,t+1} + x_{i,t} \quad \text{(for } t = 1, \ldots, k-2) \tag{90}$$

$$\sigma_{\Delta^2} = \text{std}(\Delta^2 x_i) \tag{91}$$

5. **Complexity Integration:** Combine trend and variation with patch size weighting:

$$\text{complexity}_i = \min\left(1.0, \sqrt{\beta_i^2 + \sigma_{\Delta^2}^2} \times \log(k+1)\right) \tag{92}$$

6. **Final Score:** $S_{PC} = \frac{1}{N}\sum_{i=1}^{N} \text{complexity}_i$

### D.4.4   MATHEMATICAL FORMULATION

$$S_{PC} = \begin{cases} 0.05 & \text{if } k = 1 \\ \frac{1}{N}\sum_{i=1}^{N} \min\left(1, \sqrt{\beta_i^2 + \sigma_{\Delta^2 x_i}^2} \cdot \ln(k+1)\right) & \text{if } k > 1 \end{cases} \tag{93}$$

### D.4.5   INTERPRETATION

- **High Complexity**: Patches contain rich pattern variations, trends, and periodic features
- **Low Complexity**: Patches exhibit relatively flat, monotonic, or simple signal patterns
- **Balance Point**: Neither pure noise nor overly simplistic signals

