# OpenReview forum: "The Language of Time: a Language Model Perspective on Time Series Foundation Models"
_ICLR.cc/2026/Conference — Submitted to ICLR 2026_

### Official Review · Reviewer_sGZr · 2025-10-26

**Soundness:** 2
**Presentation:** 3
**Contribution:** 3
**Rating:** 6
**Confidence:** 4

**Summary:**

The paper explores a theory-driven explanation for why time series admit transferability by casting them into a language-like paradigm: segment sequences into patches, quantize into tokens, and form a temporal vocabulary so the data can be processed by general-purpose sequence models. Methodologically, it presents (i) a channel-agnostic encoding/aggregation idea intended to produce stable token streams under multi-channel inputs; (ii) a vocabulary construction pipeline using standard clustering (e.g., K-Means); and (iii) a theory-informed narrative linking empirical observations to principles: faithful discretization (ε-covering/quantization bounds), Zipf-like natural statistics, capacity & generalization (hypothesis-space inclusion, stability bounds), and an information-theoretic view (patching as IB-style compression). Experiments report vocabulary statistics, visual analyses, and hyperparameter sensitivity (P, K) across datasets, pointing to the scalability and cross-domain applicability of this unified preprocessing-and-discretization pipeline (patch-token-vocabulary).

**Strengths:**

- Unifying perspective: converting time series into token sequences (patch→token→vocabulary) enables reuse of general sequence-model tooling across tasks/domains.
- Practical simplicity: the windowing + clustering pipeline is easy to implement and scale, and integrates with existing systems.
- Interpretability with evidence: Zipf/long-tail statistics, separability, and sensitivity analyses help explain how tokenized time behaves in distributional and geometric terms.
- Coherent theoretical narrative: the paper connects discretization, statistical structure, capacity/generalization, and IB intuitions into a single storyline.

**Weaknesses:**

1. (Section 2.2) “Channel-agnostic” should be citation-backed or down-scoped
   - Issue: The aim is to explain why transferability exists; channel-agnosticity is one facet of that explanation, not a claim to be proven here. Current wording may read as a strong conclusion.
   - Suggestion (pick one):
     (A) Citation-backed — add representative prior work that demonstrates transfer under channel changes/heterogeneity (multi-sensor/multimodal/cross-device/cross-subject), so the statement is literature-backed; or
     (B) Down-scope — rephrase it as an explanatory/inductive attribute (“a plausible factor enabling transferability”), rather than a strong empirical claim within this paper.

2. (Section 2.2.1) Quantization choice and assumptions untested
   - Issue: Only K-Means is used (Euclidean convexity, isotropic clusters); no comparisons with GMM, VQ-VAE, Product Quantization; no analysis of distance metrics (DTW, cosine, Mahalanobis).
   - Suggestion: Add ablations with alternative quantizers/metrics; report effects on Zipf fit, within-cluster variance, and downstream metrics with significance tests.

3. (Section 3.1) Intra-cluster semantics not analyzed under faithful discretization
   - Issue: The theorem bounds global ε but does not examine whether within-cluster deviations harm semantic consistency, cause boundary confusion, or break temporal smoothness.
   - Suggestion: Report correlations between within-cluster variance and downstream performance (even lightweight).

4. (Section 3.2) Misalignment between the *core question* and the *explanation*, plus lack of validation
   - Issue: §3.2 asks *why* time series follow Zipf and what it reveals about structure, yet it assumes a two-parameter GEM prior and derives Zipf—turning a causal “why” into “reproducing Zipf under a chosen prior.” There is no statistical validation that time series are GEM-like (parameter estimation, GOF/posterior checks).
   - Suggestion: Clearly distinguish explanatory hypothesis from causal proof; treat §3.2 as a generative interpretation; add minimal quantitative checks or qualify the claims accordingly.

**Questions:**

- Will you add citations that directly evidence channel-agnostic transfer (e.g., multivariate TS across sensors/devices/subjects/modalities) so that §2.2 is literature-backed?
- If not adding such citations, would you down-scope channel-agnosticity to an explanatory phrasing consistent with the paper’s “why-explanation” focus?
- How would Zipf/long-tail stats and downstream metrics change with GMM/VQ-VAE or DTW/cosine/Mahalanobis distances?
- Do intra-cluster errors correlate with downstream performance (a lightweight analysis would suffice)?
- Can you provide a minimal statistical check for a GEM-like process to better address the causal “why” in §3.2?

---

> ### Author Response · Authors · 2025-11-20
> **Responses to Comments (Part 1)**
>
> We sincerely thank you for your careful review and constructive feedback. Below we address your comments point by point.
>
> **Q1 & Q2 Channel-agnostic Concerns**
>
> Channel-agnostic (or channel-independent) models act as a widely used inductive bias in TSFM. A substantial body of recent TSFM architectures (e.g., PatchTST, TimeCLR, Moment, Chronos, and Time-MoE) are explicitly designed with channel-agnostic modeling. This trend is systematically summarized in Table III of the survey [1]. Accordingly, we follow this prevailing design choice in our framework.
>
> Ref:
>
> [1] Ye, J., Yu, Y., Zhang, W., Wang, L., Li, J., & Tsung, F. (2024). Empowering Time Series Analysis with Foundation Models: A Comprehensive Survey.
>
> **Q3 Quantization Choice and Assumptions**
>
> 1. **End-to-end prediction perspective**:
>    In our framework, the codebook index is an intermediate latent variable. Theorems A.3/A.4 show that a large class of TSFMs can be reparameterized as a “cluster selection + residual prediction” mapping. From this perspective, the continuous predictor is the main source of expressivity, while the discrete codebook only partitions the representation space. Therefore, the choice of quantizer should not fundamentally limit the model’s expressivity.
>
> | pred_len | K=16   | K=256  | K=512  |
> |----------|--------|--------|--------|
> | **96**   | 0.3206 | 0.3173 | 0.3248 |
> | **192**  | 0.3514 | 0.3503 | 0.3560 |
> | **336**  | 0.3843 | 0.3832 | 0.3833 |
> | **720**  | 0.4349 | 0.4336 | 0.4359 |
>
> The small performance differences across different K values support our view that “the codebook can be marginalized out in end-to-end tasks and has limited impact on performance.”
>
> 2. **Neutrality of statistical analysis**:
>    In empirical analyses (Zipf statistics, long-tailed structure, geometric visualizations), we intentionally adopted the simplest and most transparent quantizer (K-means) to minimize additional inductive biases and make vocabulary construction as “neutral” as possible.
>
> Therefore, the conclusion: **from the perspective of theoretical analysis, in end-to-end tasks, the codebook is a reducible latent variable, and the choice of quantization method is irrelevant to the results**; from a practical perspective, in statistical analysis, we examined results using both k-means and VQ-VAE, and the both present clear Zipf Law across different codebook size. Below is the results of VQ-VAE:
>
> | codebook size| activated cluster | Zipf α |
> |--------------|--------------|--------|
> | 16  | 16 / 16  | 1.0753 |
> | 32  | 32 / 32  | 0.9693 |
> | 256 | 210 / 256| 1.0303 |
>
> **Q4 Intra-cluster Semantics Analysis**
>
> By “intra-cluster errors,” we understand the within-cluster quantization error, i.e., the distance between a patch and its assigned centroid (or within-cluster variance). Although we do not use this exact term in the main text, Appendix C systematically analyzes the semantics of the vocabulary, the shapes of typical “temporal words,” and transition patterns between tokens, covering both intra-cluster and inter-cluster structures.
>
> Regarding the relationship to downstream tasks: in an end-to-end task, the codebook is a reducible latent variable (Thm. A.4, A.5), so its settings have little impact on downstream tasks. As in the table in Q3, varying K (which inherently changes the intra-cluster variance) leads to negligible performance fluctuation, empirically verifying our theoretical claim that intra-cluster errors are compensated by the continuous offset. In contrast, patch size induces quantization error (Thm. A.1), for which an upper bound exists that depends on P, this can be interpreted as larger P leading to larger error.

---

> > ### Author Response · Authors · 2025-11-26
> > **Responses to Comments (Part 2)**
> >
> > **Q5 GEM distribution Concerns**
> >
> > GEM(d, θ) is a Bayesian nonparametric prior defined over infinite-dimensional probability vectors, and performing rigorous hypothesis testing of its full generative mechanism is infeasible with finite samples. Therefore, in this paper, we do not attempt to “verify that the true data exactly follows GEM(d, θ).” Instead, based on empirical observation, we adopt it as an analytically tractable long-tailed generative model to explain the heavy-tailed phenomenon we observe in patch tokens. After observing that the frequency distribution of tokens exhibits a clear long-tailed characteristic, we employ GEM(d, θ) as a theoretical analysis framework and derive a Zipf-like rank–frequency relationship within it. This approach firstly observing a long-tailed phenomenon and then explaining it using Bayesian nonparametric long-tailed priors such as Pitman–Yor or GEM, is a well-established and widely adopted modeling paradigm in statistical language modeling and related fields [1–3].
> >
> > Refs:
> >
> > [1] Sudderth, E. B., & Jordan, M. I. Shared segmentation of natural scenes using dependent Pitman–Yor processes. NeurIPS 2009
> >
> > [2] Sato, I., & Nakagawa, H. Topic models with power-law using Pitman–Yor process. KDD 2010
> >
> > [3] Zhou, X., Zhang, J., & Kulis, B. Power-law graph cuts. AISTATS 2015
> >
> > We hope these clarifications and responses address the reviewer’s concerns, and we thank the reviewer again for the insightful comments.

---

### Official Review · Reviewer_4waQ · 2025-10-31

**Soundness:** 2
**Presentation:** 2
**Contribution:** 3
**Rating:** 4
**Confidence:** 4

**Summary:**

The core argument of this paper is that patch-based foundation models for time series can be regarded as a generalized form of large language models (LLMs). Its key insight lies in showing that time series patches can be quantized into “distributional tokens” exhibiting quasi-linguistic statistical properties (e.g., Zipf’s law), which explains the model’s strong cross-domain generalization ability.

From an experimental perspective, the paper demonstrates the superiority of patches over point-based representations and presents a “vocabulary” of time series. From a theoretical standpoint, it further establishes a rigorous mathematical framework—supported by a series of theorems and lemmas—to justify and validate the reasonableness and advantages of treating time series as a form of language.

**Strengths:**

1. The paper presents a novel perspective by interpreting time series through the lens of language modeling, effectively bridging the gap between time series sequences and natural language sentences. The insight presented in this paper is interesting and addresses a valuable and important problem in the design of time series foundation model encoders.
2. The argumentation is comprehensive: beyond the empirical evidence provided by experiments, the paper develops an end-to-end, logically coherent mathematical framework for the “time series as language” paradigm, drawing from representation theory, model capacity, stochastic process properties, and learning theory.

**Weaknesses:**

1. (major) The experimental and theoretical analyses in this paper are well executed; however, the connection to foundation models themselves is insufficient. This limitation is mainly reflected in two aspects:
    1. Lack of causal linkage between “distribution token” and model inference behavior:
    In the experimental section, the paper clusters time series after segmenting them into patches and observes the phenomenon of a “probability cloud,” suggesting that time series tokens are distributional tokens, with a corresponding theoretical explanation provided (Theorem A.4, Centroid–offset decomposition). However, the logical connection from this observation to the conclusion that “the model essentially operates on probability distributions” is incomplete. The paper does not provide evidence showing that the model (e.g., Transformers) can truly process such “probability clouds” as unified computational entities. At present, the “Centroid–offset” framework appears more as an effective modeling strategy for describing the fact of clustering results, rather than evidence that the model itself has evolved to manipulate probabilistic objects.
    2. Disconnected Analysis between "Quasi-Linguistic Properties" and Downstream Performance: The experimental section powerfully demonstrates the quasi-linguistic properties of time series but fails to establish a causal link between the "quality" of these properties and a model's performance on downstream tasks, such as forecasting. For instance, the paper notes a trade-off where larger patches (P=128) yield better linguistic plausibility (Zipf conformity), while smaller patches (P=32) offer superior structural fidelity (clustering quality). It remains unclear which of these characteristics is more critical for achieving high performance in downstream tasks (e.g. forecasting) and what the underlying reasons are.
2. (minor) Some of the theoretical proofs deviate considerably from practical conditions. In arguing that patch-based models have stronger representational capacity (Theorem 3.3), the proof (specifically Lemma A.3 in Appendix A.2) relies on a key assumption that the dictionary contains all possible patches of length 1. In a continuous space, this implies an uncountably infinite vocabulary. Theorem 3.2 attempts to theoretically explain the emergence of Zipf’s law by assuming that the token generation process follows a GEM distribution, from which Zipf’s law is derived. However, the paper does not provide any prior justification for why the generation process of time series “tokens” should naturally conform to a GEM distribution.

**Questions:**

1. Please refer to Weakness 1. The authors could further strengthen the paper by applying the key observations and hypotheses derived from their analyses to time series foundation models, and reporting corresponding experimental results. Such evidence would help verify whether these insights are valid in practice and whether they indeed contribute to improving the predictive performance of time series foundation models.

---

> ### Author Response · Authors · 2025-11-20
> **Responses to Comments (Part 1)**
>
> We sincerely thank you for your careful review and constructive feedback. Below we address your comments point by point.
>
> **W1.1 Causal linkage between "distributional token" and inference**
>
> We would clarify that our intention is not to claim that the model explicitly manipulates probability clouds as distinct operators in its computation graph. Rather, our claim is distributional and statistical: the prediction distribution of TSFMs is mathematically equivalent to a "discrete motif distribution + continuous offset distribution" (Thms. A.3, A.4). The causal linkage between this theoretical structure and the model’s actual behavior is established through the optimization dynamics:
>
>  -  Implicit Mechanism: Modern TSFMs typically minimize reconstruction loss (e.g., MSE). Statistically, minimizing MSE is equivalent to estimating the conditional expectation. In our framework, the "cluster center" corresponds to this expectation (the prototype), while the "offset" corresponds to the irreducible variance (the probability cloud).
>
> - Formalization: Consequently, the codebook with $K$ is implicitly modeled and ultimately marginalized out as a latent variable via the mixture density: $P(x_{t+1}|h) = \sum_{k=1}^K P(k|h) P_{\epsilon}(x_{t+1}-c_k|k,h)$ . Here, the model’s point prediction is essentially estimating the expectation of this mixture distribution.
>
> This mathematical equivalence confirms that the "distributional token" is not merely an external observation, but the intrinsic latent variable governed by the model's probabilistic inference process. The "probability cloud" phenomena observed in experiments constitute empirical evidence that the data geometry itself supports such a decomposition.
>
> **W1.2 Quasi-Linguistic Properties vs. Downstream Performance**
>
> This is a critical insight. We argue that Linguistic Plausibility (Generalization) and Structural Fidelity (Precision) play complementary, yet competing roles.
>
> * Foundation (Generalization): The adherence to Zipf's law (observed strongly at P=128) suggests that time series possess a language-like compressible structure, which is the prerequisite for the "foundation model" paradigm to work (enabling abstraction and transfer).
> * Precision (Accuracy): However, overly large patches (like P=128) smooth out local variations, increasing the quantization error bound (Theorem A.1, $\epsilon \propto \sqrt{P}\$). Conversely, P=1 captures details but lacks structure.
>
> Therefore, the optimal performance is found at a "sweet spot" (e.g., P=32) where the model benefits from both strong quasi-linguistic properties (validating the language paradigm) and sufficient local semantic fidelity. The experimental results below confirm this: $P=32$ consistently outperforms the extremes ($P=1$ or very large $P$), demonstrating that downstream success relies on balancing linguistic abstraction with local detail preservation.
>
> Here, we provide supporting experimental results on downstream tasks:
>
> | Dataset   | pred_len | MSE@p16 | MSE@p32 | MSE@p64 | MSE@p128 |
> |-----------|----------|---------|---------|---------|----------|
> | ETTm1     | 96       | 0.3142  | 0.3173  | 0.3429  | 0.3977   |
> | ETTm1     | 720      | 0.4491  | 0.4336  | 0.4565  | 0.4938   |
> | Weather   | 96       | 0.1697  | 0.1602  | 0.1740  | 0.1854   |
> | Weather   | 720      | 0.3260  | 0.3244  | 0.3318  | 0.3365   |
>
> We also provide the Pattern Complexity from P=1 to 2048:
>
> |  P | 1 | 16 | 32 | 64 | 128 | 256 |512|1024|2048|
> |------------|---|----|----|----|-----|-----|-----|-----|-----|
> | Pattern  Complexity | 0.050 | 0.192 | 0.226 | 0.258 | 0.245 | 0.254 |0.238|0.236|0.230
>
> These results also indicate that P has an optimal value range, despite the range different from in downstream conclusions due to different metrics, which are consistent with our analysis.
>
> > Our primary goal in this work is not to propose yet another architecture that directly improves state-of-the-art forecasting metrics, but to establish a principled bridge between time-series patch models and language modeling. This bridge provides a theoretical and empirical anchor that clarifies why patch-based TSFMs behave in an LLM-like way, so that future TSFMs can systematically leverage design principles and successes from large language models to improve downstream performance.

---

> ### Author Response · Authors · 2025-11-23
> **Responses to Comments (Part 2)**
>
> **W2 Theoretical Assumptions**
>
> - Infinite Dictionary. For Lemma A.3 (Appendix A.2.), our aim is to show that patch-based models subsume point-based ones via expressivity upper bounds. The “infinite codebook” is a standard theoretical device to establish such asymptotic bounds, an idealization we explicitly acknowledge in Remark A.4, not a practical design. Crucially, our approach does not depend on this limit: Thm. A.1 provides a finite-codebook counterpart, bounding quantization error $\epsilon$ via finite codebook size K (specifically, $K(\epsilon) = \mathcal{O}(\epsilon^{-(P-1)})$) in real systems. Together, Lemma A.3 and Thm. A.1 form a complete picture: the former gives an ideal expressivity ceiling; the latter controls real-world error.
>
> - GEM Distribution."Empirical Observation $\to$ Bayesian Modeling" pipeline, which is standard in machine learning. Since we empirically observe that patch frequencies follow a heavy-tailed distribution (long-tail), the GEM is the rigorous nonparametric prior to model this generative mechanism [1-3]. We do not claim the physical world is a GEM process, but that GEM is the statistically optimal proxy for modeling the observed rank-frequency dynamics.
>
> Refs:
>
> [1] Sudderth, E. B., & Jordan, M. I. Shared segmentation of natural scenes using dependent Pitman–Yor processes. NeurIPS 2009
>
> [2] Sato, I., & Nakagawa, H. Topic models with power-law using Pitman–Yor process. KDD 2010
>
> [3] Zhou, X., Zhang, J., & Kulis, B. Power-law graph cuts. AISTATS 2015
>
>
> We hope these clarifications and responses address the reviewer’s concerns, and we thank the reviewer again for the insightful comments.

---

### Official Review · Reviewer_XNEz · 2025-10-31

**Soundness:** 3
**Presentation:** 3
**Contribution:** 2
**Rating:** 6
**Confidence:** 2

**Summary:**

The paper reframes time series foundation models as language models over patch-level “distributional tokens.” Concretely, each short segment is assigned to a motif cluster and the model predicts a continuous offset around that centroid, yielding a cluster-and-offset pipeline that explains transfer across domains. Empirically, the authors compile a large “time-series vocabulary” across 37 heterogeneous datasets and show language-like statistics. Theoretically, they provide quantization/coverage error bounds and a capacity hierarchy showing that the hypothesis space of pointwise models is strictly contained in that of patch models.

**Strengths:**

1. The paper tackles an engaging problem and advances a theoretical account of why TSFMs can transfer and generalize across domains.
2. The study substantiates its claims with empirical evidence on diverse real-world time-series corpora, using these datasets to test the central assumptions of the proposed framework.

**Weaknesses:**

1. The empirical validity appears sensitive to tokenization design, including codebook size, patch length, normalization, and clustering, leaving questions about the robustness and cross-domain stability of these hyperparameters.
2. The theoretical guarantees rely on smoothness and boundedness assumptions; while strong assumptions are acceptable in a theoretical treatment, their realism under non-stationarity and abrupt regime shifts remains insufficiently examined.

**Questions:**

1. When deploying to a new domain with shifted distributions, how should the codebook evolve?
2. How specifically do patch length and codebook size trade bias and variance in the cluster-and-offset scheme.

---

> ### Author Response · Authors · 2025-11-20
> **Responses to Comments (Part 1)**
>
> We sincerely thank you for your careful review and constructive feedback. Below we address your comments point by point.
>
> **W1 Robustness to Codebook Generation**
>
> **Codebook Size (K)**
> Cluster selection + residual prediction (Thm. A.3 / A.4) shows that many TSFM architectures can be reparameterized as “first select a cluster center, then predict a continuous offset.” Under this view, K is not a fundamental structural bottleneck for model capacity, the codebook with K is implicitly modeled and ultimately marginalized out as a latent variable via the mixture density  $P(x_{t+1}|h) = \sum_{k=1}^K P(k|h) P_{\epsilon}(x_{t+1}-c_k|k,h)$.
>
> We supplement results for Tokenized PatchTST across different values of $K$:
>
> | pred_len | K=16   | K=256  | K=512  |
> |----------|--------|--------|--------|
> | **96**   | 0.3206 | 0.3173 | 0.3248 |
> | **192**  | 0.3514 | 0.3503 | 0.3560 |
> | **336**  | 0.3843 | 0.3832 | 0.3833 |
> | **720**  | 0.4349 | 0.4336 | 0.4359 |
>
> Performance remains nearly unchanged across a 32× variation in K, indicating downstream tasks are insensitive.The core macroscopic statistical structures we emphasize: Zipf-like heavy-tailed token frequencies and long-tailed motif distributions remain stable across K (Figure 5).
>
> **Patch Length (P)**
>   P directly governs how local temporal context is preserved: too small a P approaches pointwise prediction and loses local dynamics (Lemma A.3); too large a P increases quantization error and token sparsity (Thm. A.1, A.7, A.8). Thus, P embodies a bias–variance trade-off. Crucially, the language-like statistical properties remain stable across different P:
>
>   **Stability of Zipf exponents w.r.t. P**
>
> | Dataset  | P=16 | P=32 | P=64 | P=128 |
> |----------|------|------|------|-------|
> | Weather  | 0.87 | 0.93 | 0.74 | 0.87  |
> | Energy   | 0.95 | 0.94 | 0.87 | 0.77  |
> | Traffic  | 0.97 | 0.90 | 0.85 | 0.76  |
>
>   All exponents fall within or near the typical Zipfian range(~1 [1]), supporting our statements.
>
>   **2-gram sparsity**
>
> | Dataset  | P=16   | P=32   | P=64   |
> |----------|--------|--------|--------|
> | Weather  | 0.9726 | 0.9860 | 0.9932 |
> | Energy   | 0.9858 | 0.9894 | 0.9941 |
> | Traffic  | 0.9818 | 0.9869 | 0.9933 |
>
>   These results confirm that while P affects downstream performance, key language-like phenomena persist consistently across P. Empirically, we find $P \in [16, 64]$ to be the most stable range across domains. The core empirical findings (Zipf, n-gram sparsity, cluster geometry) do not rely on any of these assumptions.
>
> **Normalization and Clustering**
>   Normalization is a standard design choice in time series. We observe consistent language-like structures under various normalization schemes. Regarding clustering, our theoretical analysis applies to general quantizers (Thm. A.1, A.3). We adopt K-means for its simplicity, interpretability, and alignment with existing TSFM literature. We also tested alternatives (e.g., VQ-VAE), and the core statistical patterns remain unchanged.
>
> Ref:
>
> [1] Newman, M. E. J. “Power Laws, Pareto Distributions and Zipf’s Law.” Contemporary Physics, vol. 46, no. 5, 2005, pp
>
> **W2 Theoretical Guarantees**
>
> We clarify that these assumptions are theoretical sufficient conditions for Thm. 3.4, not strict physical constraints. These assuptions are also widely applied and well-established in time series and stochastic processes, we follow these settings [1-2]. In practice, our design aligns data with these conditions: Patching acts as a local smoothing operator to mitigate abrupt shifts, while Instance Normalization stabilizes dynamic ranges to satisfy boundedness with high probability. We quantitatively verified robustness to non-stationarity via $\beta$-mixing hypothesis testing (L=512～12288). Results confirm that patching preserves the mixing property with strictly controlled deviations (0.05～0.14), where violation rates decrease in longer sequences. This indicates that observed fluctuations stem primarily from estimation variance rather than structural failure, validating the framework's applicability to real-world dynamics.
>
> Refs:
>
> [1] Bradley, R. C. . Basic properties of strong mixing conditions. A survey and some open questions. Probability surveys, 2005,
>
> [2] Yu, B.. Rates of convergence for empirical processes of stationary mixing sequences. The Annals of Probability, 1994.

---

> ### Author Response · Authors · 2025-11-23
> **Responses to Comments (Part 2)**
>
> **Q1 Deployment Concerns**
>
> Whether the codebook needs to be updated depends on whether we explicitly adopt a “cluster selection + residual prediction” architecture.
>
> - Implicit codebooks:
>   Serve as latent variables marginalized out during inference. No explicit updates are needed for transfer as the model adapts via continuous parameters.
> - Explicit codebooks :
>   We recommend a dynamic codebook with bounded capacity (e.g., VQ-VAE). This allows retaining pretrained high-frequency motifs while activating new codes on-demand, enabling a lightweight hot-update mechanism for emerging patterns.
>
> **Q2 Trade-off of $K$ and $P$**
>
> - Patch length
>   Too small implies loss of local context, near pointwise modeling; too large implies sparser tokens, larger quantization error. Thus, P trades off context preservation vs. sparsification/quantization error.
> - Codebook size
>   Too small implies distinct patterns merged, coarse quantization; too large implies few samples per cluster, unstable offset/transition estimates. Thus,  K  trades off representation fidelity vs. estimation stability.
>
> We hope these clarifications and responses address the reviewer’s concerns, and we thank the reviewer again for the insightful comments.

---

### Official Review · Reviewer_QRvq · 2025-10-31

**Soundness:** 3
**Presentation:** 4
**Contribution:** 4
**Rating:** 6
**Confidence:** 4

**Summary:**

Current Time Series Foundation Models have demonstrated strong practical performance and maintain generalization and robustness across domains; however, their interpretability remains underexplored. This paper takes the perspective of treating time series as a “language”, borrowing linguistic insights—particularly distributional patterns from natural language—to explain the effectiveness of Time Series Foundation Models. The authors conduct extensive comparative experiments to analyze how factors such as patch size and cluster size influence the linguistic properties of time series and further propose a theoretical framework to support their hypothesis.

**Strengths:**

Time series in foundation models are segmented into patches based on stride and patch size, analogous to tokens in LLMs. The paper identifies this resemblance and attempts to transfer linguistic distribution laws—such as Zipf’s law—from natural language modeling to time-series modeling, providing a novel and insightful analogy that helps explain model effectiveness.

**Weaknesses:**

While the analogy-based narrative and theoretical construction are relatively coherent, several aspects are counterintuitive, and some experimental evidence appears insufficient to fully support the claims.

**Questions:**

1. Vocabulary Size Limitation:
   Modern LLMs typically have vocabularies in the hundreds of thousands (e.g., Qwen3: ~151,936 tokens), where Zipf’s law manifests as a power-law distribution explaining language expressiveness. In contrast, this paper adopts a cluster size of 32, meaning only 32 “time-series tokens.” Such a small vocabulary weakens the explanatory force of Zipf-like reasoning.


$ P(X = k) = \frac{1/k^s}{H_{N,s}}, \quad k = 1, 2, \dots, N $
$ H_{N,s} = \sum_{n=1}^{N} \frac{1}{n^s} $

When N goes to Inf:
$ P(X = k) = \frac{1/k^s}{\zeta(s)}, \qquad s > 1 $


2. Blurry Cluster Boundaries:
   The paper notes that latent time-series representations follow probability density distributions, leading to fuzzy boundaries between clustered tokens—visible in visualization results. Additional experiments are needed to explore how this probabilistic nature impacts model performance. For instance, if all time series within a cluster are treated as one patch token, what are the trade-offs in downstream tasks?

3. Fixed Patch Size Issue:
   The study uses a fixed patch size of 32 across time series with different frequencies and sampling rates. This design strongly affects the “time-series as language” interpretation, but the paper does not analyze its implications—especially concerning frequency scalability in foundation models.

4. Generalization Across Settings:
   The authors fix both patch size and cluster number to 32 and derive conclusions from this setup. It is unclear whether these conclusions hold under different patch or cluster configurations. More analysis or experiments are needed to validate generalization.

5. Insufficient Explanation of Cluster Centers:
   In Appendix (page 27), the authors visualize 32 cluster centers, implying these represent fundamental time-series patterns that can cover a broad range of temporal dynamics. However, the explanation is insufficient—particularly regarding how these 32 patterns capture the diversity of time-series structures, similar to how DCT basis components reconstruct images in JPEG compression.

6. Concerns on Realism of Theoretical Assumptions
Theorems assume β-mixing, GEM distribution, stability conditions, etc. Do real time-series datasets satisfy these assumptions? Have the authors validated β-mixing or GEM-like process assumptions empirically? Some TS domains (finance, climate, sensors) may not have β-mixing or stationary structure.

---

> ### Author Response · Authors · 2025-11-20
> **Responses to Comments (Part 1)**
>
> We sincerely thank you for your careful review and constructive feedback. Below we address your comments point by point.
>
> **Q1 Vocabulary Size & Validity of Zipf's Law**
>
> The “vocabulary” in language models differs fundamentally from the role of K in time series. Here, tokens are data-driven statistical prototypes from a continuous motif manifold rather than predefined symbols. Thm. A.3 and A.4 show that most TSFM architectures can be reparameterized into a “cluster selection + residual prediction’’ mechanism. Thus, the codebook with K is implicitly modeled and ultimately marginalized out as a latent variable via the mixture density  $P(x_{t+1}|h) = \sum_{k=1}^K P(k|h) P_{\epsilon}(x_{t+1}-c_k|k,h)$: varying K refines latent-space quantization but does not constrain the predictor’s hypothesis space. Consequently, K governs manifold granularity rather than imposing a symbolic bottleneck as in language models.
>
> Regarding Zipf’s law, we do not rely on larger K. In linguistics, a rank–frequency exponent in [0.8, 1.2] is commonly regarded as Zipfian [1], and all our empirical exponents lie well within this range (Fig. 5A).
>
> Ref:
>
> [1] Newman, M. E. J. “Power Laws, Pareto Distributions and Zipf’s Law.” Contemporary Physics, vol. 46, no. 5, 2005, pp
>
> **Q2 Impact of Blurry Boundaries on Performance**
>
> Boundary blurriness supports the distributional token hypothesis, which naturally arises when quantizing a continuous dynamical manifold, reflecting its underlying probability density distribution. As Lemma A.2, K-means induces a Voronoi partition in patch space. Since patches encode overlapping local dynamics, embeddings frequently lie near Voronoi boundaries, approximating regions of high uncertainty in the latent distribution. Increasing K refines local structure; decreasing K yields coarser but clearer partitions. To assess downstream impact where all patches in a cluster are treated as one patch token, we evaluated a tokenized PatchTST model (each patch mapped to a cluster ID):
>
> | pred\_len | K=16   | K=256  | K=512  |
> | ----------- | -------- | -------- | -------- |
> | 96        | 0.3206 | 0.3173 | 0.3248 |
> | 192       | 0.3514 | 0.3503 | 0.3560 |
> | 336       | 0.3843 | 0.3832 | 0.3833 |
> | 720       | 0.4349 | 0.4336 | 0.4359 |
>
> Performance varies minimally across wide K ranges, indicating that blurry boundaries limitedly impact end-to-end forecasting. Regarding your question:
> For end-to-end tasks, K selection has negligible effects as K is marginalized out as a latent variable. For analysis tasks (e.g., clustering interpretation), larger K preserves finer semantics but may amplify boundary uncertainties.
>
> **Q3 & Q4 Generalization & Patch Size Robustness**
>
> Unlike K, which is analyzed above, the patch length P determines temporal aggregation. Too small P reduces modeling to pointwised too large P increases quantization error. Thus, P trades off context retention and discretization error, as in Lemma A.3 and Thm. 3.5. Empirically, P∈[16,64] is robust. We measured linguistic-statistic stability across P:
>
> **Zipf exponents**
>
> | Dataset | P=16 | P=32 | P=64 | P=128 |
> | ------- | ---- | ---- | ---- | ----- |
> | Weather | 0.87 | 0.93 | 0.74 | 0.87  |
> | Energy  | 0.95 | 0.94 | 0.87 | 0.77  |
> | Traffic | 0.97 | 0.90 | 0.85 | 0.76  |
>
> **2-gram sparsity**
>
> | Dataset | P=16   | P=32   | P=64   |
> | ------- | ------ | ------ | ------ |
> | Weather | 0.9726 | 0.9860 | 0.9932 |
> | Energy  | 0.9858 | 0.9894 | 0.9941 |
> | Traffic | 0.9818 | 0.9869 | 0.9933 |
>
> Across P, we consistently observe heavy-tailed distributions and highly sparse 2-grams, showing that language-like properties stable. Prediction performance is more sensitive to P, and P=32 provides a practical balance. Our datasets span sampling intervals from seconds to months, fixing timesteps per patch implicitly evaluates vocabulary robustness across time scales.
>
> **Q5 Nature of Cluster Centers (Prototypes vs. DCT)**
>
> We have expanded Appendix C.2. Fundamentally, our approach differs from DCT: while DCT relies on the linear superposition of analytical basis functions to reconstruct signals, our centers are data-driven prototypes used to match time-series subsequences to their nearest statistical patterns. Instead of additive reconstruction, each center acts as a Voronoi centroid aggregating a family of topologically similar motifs. Thus, the diversity of temporal dynamics is captured through the temporal composition of these discrete prototypes, rather than linear combination.

---

> ### Author Response · Authors · 2025-11-23
> **Responses to Comments (Part 2)**
>
> **Q6 Theoretical Assumptions**
>
> We do not claim that real-world time series strictly satisfy these theoretical assumptions. However, we have added relevant literature and conducted hypothesis testing, which confirm that these assumptions are empirically valid within our context:
>
> * We employ $\beta$-mixing as a regularity condition on time series and stochastic processes, which is a well-established and widely-applied assumption [1-2]. Because patching smooths noise and dampens long-range dependence, this assumption aligns well with empirical n-gram statistics. We emphasize that $\beta$-mixing serves as a sufficient condition for our theoretical framework, rather than a mandatory assumption regarding the underlying physical dynamics. We quantitatively verified robustness to non-stationarity via mixing hypothesis testing (L=512～12288). Results confirm that patching preserves the mixing property with strictly controlled deviations (0.05～0.14), where violation rates decrease in longer sequences
>
> * The GEM prior (Thm. 3.2) is adopted for its capacity to naturally model strong heavy tails. As a well-established Bayesian nonparametric framework [3–5], its sorted mass exhibits Zipf-like behavior, paralleling standard practices in NLP. We clarify that the GEM serves as an explanatory model for the observed statistics, rather than a rigid generative mechanism.
>
> Refs:
>
> [1] Bradley, R. C. Basic properties of strong mixing conditions. A survey and some open questions. Probability surveys, 2005,
>
> [2] Yu, B.. Rates of convergence for empirical processes of stationary mixing sequences. The Annals of Probability, 1994.
>
> [3] Sudderth, E. B. et.al. Shared segmentation of natural scenes using dependent Pitman–Yor processes. NeurIPS 2009
>
> [4] Sato, I., et.al. Topic models with power-law using Pitman–Yor process. KDD 2010
>
> [5] Zhou, X., et.al. Power-law graph cuts. AISTATS 2015
>
> We hope these clarifications and responses address the reviewer’s concerns, and we thank the reviewer again for the insightful comments.

---

### Public Comment · ~Qideng_Tang1 · 2025-11-15
**Why Does a 32-Token Vocabulary Capture Natural Language–Like Distributions in Time Series?**

Thank you for your excellent, outstanding work and insightful findings. I have a small question: Natural language vocabularies are typically very large (comprising thousands of words), so why does a time series require only a vocabulary of 32 tokens (i.e., 32 clusters from k-means) to exhibit distributional characteristics similar to those of natural language vocabularies?

---

> ### Author Response · Authors · 2025-11-20
> **Response to public comment**
>
> Thank you for the insightful question. Actually, it is independent between vocabulary size and if it exhibits language-like properties:
>
> - Zipf’s Law is about shape, not size. Zipf's law characterizes the inequality of frequencies (a few dominant patterns vs. a long tail). This power-law structure is scale-invariant and manifests clearly regardless of whether the vocabulary size is 32 or 30,000.
>
> - Tokens are Prototypes on a Manifold. Unlike distinct semantic words, time series patches lie on a continuous manifold. Fundamental dynamical patterns are limited. $K$ captures this "skeleton," while diversity is modeled by the continuous offsets rather than distinct token IDs.
>
> - Latent Anchor for Alignment. Theoretically, the vocabulary serves as a reducible latent variable derived from factorizing the joint probability. This latent variable acts as the "anchor point" that aligns TSFMs with the LLM paradigm. Since this variable is effectively marginalized out during inference (via the mixture density), the specific vocabulary size does not bottleneck end-to-end performance.

---

> > ### Public Comment · ~Qideng_Tang1 · 2025-11-28
> >
> > Thank you for your response and this is an excellent piece of work.

---

### Author Response · Authors · 2025-11-25
**Common Concerns Summary (Part 1)**

We thank all reviewers for their insightful and constructive feedback. Since several reviewers raised overlapping questions regarding the validity of the vocabulary analogy, hyperparameter sensitivity (Patch Length $P$ & Cluster Size $K$), and the realism of theoretical assumptions, we summarize our clarifications here to provide a unified perspective.

> 1. Why does a small $K$ (e.g., 32) exhibit language-like properties?

**Common Concerns** How can a time series vocabulary with only $K=32$ capture "language-like" laws (e.g., Zipf's Law)? Does this small size invalidate the analogy? How $K$ affects the downstream tasks?

**Response** We clarify that the small vocabulary size ($K$) neither invalidates the language analogy nor bottlenecks downstream performance. This is due to three key factors:

- Scale Invariance of Zipf’s Law: Zipf's law characterizes the inequality of frequency distributions (the shape), not the absolute number of unique elements. Our empirical results verify that the power-law structure ($\alpha \approx 1.0$) is scale-invariant and manifests consistently.

- Prototypes on a Manifold vs. Discrete Symbols: Unlike NLP where tokens are distinct semantic units, time series patches lie on a continuous manifold. In our framework, the $K$ tokens act as discretized prototypes (anchors) that capture the diverse "skeleton" of dynamics, while the continuous offsets model the specific variations. Thus, the total information is not compressed into just 32 integers but is represented by the "Cluster Center + Continuous Offset" pair.

- $K$ as a Latent Variable in Downstream Tasks: Theorems A.3 and A.4 demonstrate that many TSFMs effectively model a mixture density distribution. In this view, the cluster assignment is a latent variable that is marginalized out during inference. Consequently, $K$ determines the granularity of the latent space but does not constrain the predictor's hypothesis space. We evaluated a tokenized PatchTST model (each patch mapped to a cluster ID), showing that the downstream prediction performance keeps stable with varying $K$ :
| pred\_len | K=16   | K=256  | K=512  |
| ----------- | -------- | -------- | -------- |
| 96        | 0.3206 | 0.3173 | 0.3248 |
| 192       | 0.3514 | 0.3503 | 0.3560 |
| 336       | 0.3843 | 0.3832 | 0.3833 |
| 720       | 0.4349 | 0.4336 | 0.4359 |

Here, we would emphasize: Although $K$ and the codebook serve as marginalized latent variables in end-to-end tasks, this probabilistic factorization is fundamental. The vocabulary acts as the theoretical anchor that formally aligns continuous time series with the discrete LLM paradigm. This alignment allows us to characterize TSFMs as generalized language models, thereby theoretically explaining how they inherit the scaling laws and transfer capabilities of LLMs, resolving the paradox of their counter-intuitive effectiveness.

> 2. The Trade-off of Patch Length ($P$)

**Common Concerns** How does Patch Length ($P$) affect the "language" properties and downstream performance?

**Response** We identify a fundamental trade-off governed by $P$. Crucially, we first emphasize that while varying $P$ changes the granularity of the vocabulary, the core linguistic statistical properties remain robust across a wide range of $P$:

**Zipf exponents**

| Dataset | P=16 | P=32 | P=64 | P=128 |
| ------- | ---- | ---- | ---- | ----- |
| Weather | 0.87 | 0.93 | 0.74 | 0.87  |
| Energy  | 0.95 | 0.94 | 0.87 | 0.77  |
| Traffic | 0.97 | 0.90 | 0.85 | 0.76  |

**2-gram sparsity**

| Dataset | P=16   | P=32   | P=64   |
| ------- | ------ | ------ | ------ |
| Weather | 0.9726 | 0.9860 | 0.9932 |
| Energy  | 0.9858 | 0.9894 | 0.9941 |
| Traffic | 0.9818 | 0.9869 | 0.9933 |

Across $P$, we consistently observe heavy-tailed distributions and highly sparse 2-grams, showing that language-like properties stable. Prediction performance is more sensitive to $P$. For downstream forecasting, $P$ balances two competing factors:

- Linguistic Plausibility (Generalization): Larger $P$ (e.g., 128) captures longer dependencies and exhibits stronger Zipfian traits, validating the "Foundation Model" paradigm of compressibility and abstraction.

- Structural Fidelity (Precision): Smaller $P$ minimizes quantization error (Theorem A.1) but risks losing semantic structure (degenerating to pointwise noise).

 - The Sweet Spot: Our results show that $P=32$ (or roughly $[16, 64]$) strikes the optimal balance. It is large enough to form statistically significant "words" (language-like structure) but small enough to preserve local dynamics for precise forecasting. This explains why the "most linguistic" setting isn't always the "most accurate" setting—the model needs both structure and detail.

---

> ### Author Response · Authors · 2025-11-25
> **Common Concerns Summary (Part 2)**
>
> > 3. Realism of Theoretical Assumptions
>
> **Common Concerns** Real-world time series are often non-stationary and may not strictly follow $\beta$-mixing or GEM distributions. Do these assumptions hold?
>
> **Response** We clarify that these assumptions serve as theoretical sufficient conditions and explanatory proxies, rather than rigid physical constraints. We emphasize that employing $\beta$-mixing as a regularity condition and GEM as a generative prior are well-established practices grounded in mature mathematical frameworks [1-5]. Our work aligns with these standard theoretical paradigms to ensure rigorousness.
>
> - GEM as a Standard Bayesian Prior: We do not claim that the physical generation process of time series strictly follows a GEM. Rather, given the empirically observed heavy-tailed distribution, the GEM distribution is the widely adopted Bayesian nonparametric prior to model such rank-frequency dynamics [1-3]. Adopting GEM to explain power-law phenomena is a standard practice in statistical language modeling and network analysis.
>
> - Patching Enhances Mixing Stability: While raw time series can be non-stationary, the patching operation acts as a local smoothing operator, which dampens high-frequency noise and mitigates abrupt distributional shifts. The $\beta$-mixing assumption is a classic regularity condition in time series theory [4-5]. Our hypothesis testing confirms that patched sequences preserve this property with strictly controlled deviations.
>
> Refs:
>
> [1] Sudderth, E. B. et.al. Shared segmentation of natural scenes using dependent Pitman–Yor processes. NeurIPS 2009
>
> [2] Sato, I., et.al. Topic models with power-law using Pitman–Yor process. KDD 2010
>
> [3] Zhou, X., et.al. Power-law graph cuts. AISTATS 2015
>
> [4] Bradley, R. C. Basic properties of strong mixing conditions. A survey and some open questions. Probability surveys, 2005,
>
> [5] Yu, B.. Rates of convergence for empirical processes of stationary mixing sequences. The Annals of Probability, 1994.
>
> ***
> We sincerely thank all reviewers for their detailed feedback and the public commenter for the insightful discussion. We hope our responses have resolved your questions. As the discussion period progresses, we are eager to hear your feedback and feel free to let us know if any further clarification is needed.

---

### Author Response · Authors · 2025-11-26
**Following up on our Responses**

Dear Reviewers,

We would like to kindly follow up on our responses posted. We have made every effort to address your concerns and questions, including common concerns summary and point-by-point responses.

As the discussion period progresses, we are eager to engage in further discussion to clarify any remaining uncertainties. We would greatly appreciate your feedback on whether our response has resolved your concerns.

Thank you for your time and insightful review feedback.

Best regards,

Authors

---

### Meta-Review · Area_Chair_izn1 · 2026-01-13

**Summary:**

The paper carries out a well-executed investigation on what underlies a wide collection of datasets, and backs it with broad corpus statistics and a reasonably unified theoretical narrative (quantization, capacity inclusion, stability/regularity). The reviews converge on “interesting and potentially clarifying,” with the main hesitation being whether the linguistic/statistical story is robust and whether it truly explains TSFM behavior rather than offering a post-hoc analogy.

While being a comprehensive study, the work doesn’t really support the stronger message the paper wants to convey, namely, that foundation time-series models generalize across domains because they operate over these distributional tokens. The missing piece that happens to be central to the narrative is the model (pointed out by reviewer 4waQ as well). There is little to no direct evidence about what a TSFM represents, attends to, or uses for prediction, nor how these proposed linguistic measures relate to transfer performance. Without experiments that probe the model (with any of the TSFMs, Chronos, TimesFM, Morai, etc.) or demonstrate a concrete data–model mechanism (beyond post-hoc token statistics), the central claim about cross-domain generalizability reads more like an analogy than a grounded explanation. This is where I'd recommend the authors focusing on in the next revision.

**Reviewer Concerns:**

The rebuttal meaningfully addresses the most concrete concerns: it clarifies the role of K (as a latent prototype index rather than an NLP-style bottleneck), adds ablations suggesting downstream insensitivity to large swings in K, and sharpens the trade-off in patch length P, including evidence that language-like statistics persist across a range while forecasting has a sweet spot. The remaining gap is mostly about claims and validation.

**Reviewer Scores:**

All reviewers are likely to maintain their scores.

---

### Decision · Program_Chairs · 2026-01-26

Reject